# `POP-3D`: Open-Vocabulary 3D Occupancy Prediction from Images

**Antonin Vobecky**[1,2,3]    **Oriane Siméoni**[1]    **David Hurych**[1]    **Spyros Gidaris**[1]

**Andrei Bursuc**[1]    **Patrick Pérez**[1]    **Josef Sivic**[2]

[1] valeo.ai, Paris, France    [2] CIIRC CTU in Prague    [3] FEE CTU in Prague

## Abstract

We describe an approach to predict open-vocabulary 3D semantic voxel occupancy map from input 2D images with the objective of enabling 3D grounding, segmentation and retrieval of free-form language queries. This is a challenging problem because of the 2D-3D ambiguity and the open-vocabulary nature of the target tasks, where obtaining annotated training data in 3D is difficult. The contributions of this work are three-fold. First, we design a new model architecture for open-vocabulary 3D semantic occupancy prediction. The architecture consists of a 2D-3D encoder together with occupancy prediction and 3D-language heads. The output is a dense voxel map of 3D grounded language embeddings enabling a range of open-vocabulary tasks. Second, we develop a *tri-modal* self-supervised learning algorithm that leverages three modalities: (i) images, (ii) language and (iii) LiDAR point clouds, and enables training the proposed architecture using a strong pre-trained vision-language model without the need for any 3D manual language annotations. Finally, we demonstrate quantitatively the strengths of the proposed model on several open-vocabulary tasks: Zero-shot 3D semantic segmentation using existing datasets; 3D grounding and retrieval of free-form language queries, using a small dataset that we propose as an extension of nuScenes. You can find the project page here https://vobecant.github.io/POP3D.

## 1 Introduction

The detailed analysis of 3D environments –both geometrically and semantically– is a fundamental perception brick in many applications, from augmented reality to autonomous robots and vehicles. It is usually conducted with cameras and/or laser scanners (LiDAR). In its most complete version, called *semantic 3D occupancy* prediction, this analysis amounts to labelling each voxel of the perceived volume as occupied by a certain class of object or empty. This is extremely challenging since both cameras and LiDAR only capture information about visible surfaces, which may be projected from 3D into 2D without the loss of information, but not for every point in the 3D space. This one extra dimension makes prediction arduous and hugely complicates the manual annotation task.

Recent works, e.g., [26], propose to leverage manually-annotated LiDAR data to produce a partial annotation of the 3D occupancy space. However, relying on manual semantic annotation of point clouds remains difficult to scale, even if sparse, and limits the learned representation to encode solely a closed vocabulary, i.e., a limited predefined set of classes. In this work, we tackle these challenges and propose an open-vocabulary approach to 3D semantic occupancy prediction that relies only on

---

[2]Czech Institute of Informatics, Robotics and Cybernetics at the Czech Technical University in Prague

37th Conference on Neural Information Processing Systems (NeurIPS 2023).

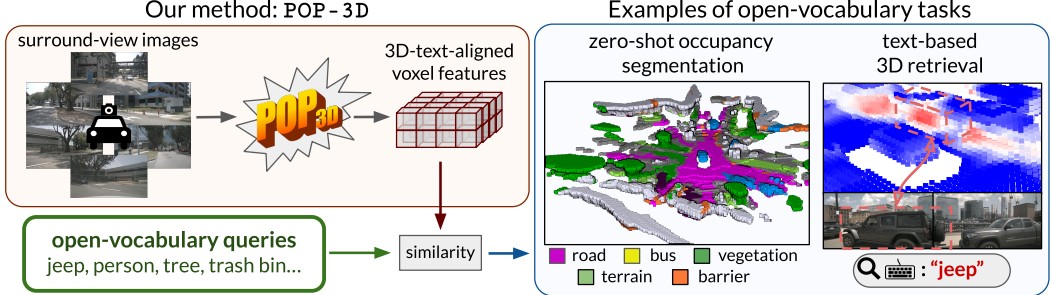

Figure 1: **Overview of the proposed method.** Provided only with surround-view images as input, our model called POP-3D produces a voxel grid of 3D text-aligned features that support open-vocabulary downstream tasks such as zero-shot occupancy segmentation or text-based grounding and retrieval.

unlabeled image-LiDAR data for training. In addition, our model uses only camera inputs at run time, bypassing altogether the need for expensive dense LiDAR sensor, in contrast with most 3D semantic perception systems (whether at point or voxel level).

To this end, we harness the progress made recently in supervised 3D occupancy prediction [26] and in language-image alignment [64], within a two-head image-only model that can be trained with aligned image-LiDAR raw data. Leveraging sparse 3D occupancy information that LiDAR scans provide for free, we first train a class-agnostic occupancy prediction head. Using this same LiDAR information along with pre-trained language-aligned visual features at the corresponding locations in images, we jointly train a second head that predicts the same type of features at the 3D voxel level. At run time, these features can be probed from text prompts to get open-vocabulary semantic segmentation of voxels that are predicted as occupied (Fig. 1). To assess the effectiveness of our method for semantic 3D occupancy prediction, we introduce a novel evaluation protocol specifically tailored to this task. Through evaluation with this protocol on autonomous driving data, our method is shown to achieve a strong performance relative to the fully-supervised approach.

In a nutshell, we attack the difficult problem of 3D semantic occupancy prediction with the lightest possible requirements: no manual annotation of the training data, no pre-defined semantic vocabulary, and no recourse to LiDAR readings at run time. As a result, the proposed image-only 3D semantic occupancy model named POP-3D (for o**P**en-vocabulary **O**ccupancy **P**rediction in **3D**) provides training data scalability and operational versatility, while opening up new understanding capabilities for autonomous systems through language-driven scene perception.

## 2 Related work

**Semantic 3D occupancy prediction.** Automatic understanding of the 3D geometry and semantics of a scene has been traditionally enabled through high precision LiDAR sensors and corresponding architectures. 3D semantic segmentation, i.e., point-level classification of a point cloud, can be addressed with different types of transformations of the point cloud: point-based, directly operating on the three-dimensional points [45, 46, 53], and projection-based, operating on a different representation, *e.g.*, two-dimensional images [57, 32, 8] or three-dimensional voxel representations [61, 65, 52, 18]. However, they produce predictions as sparse as the LiDAR point cloud offering an incomplete understanding of the full scene. Semantic scene completion [50] aims for dense inference of 3D geometry and semantics of objects and surfaces within a given extent, typically leveraging rich geometry information at the input extracted from depth [16, 35], occupancy grids [58, 49], point clouds [48], or a mix of modalities, e.g., RGBD [11, 17]. In this line, MonoScene [12] is the first camera-based method to produce dense semantic occupancy predictions from a single image by projecting image features into 3D voxels by optical ray intersection. Recent progress in multi-camera Bird's-Eye-View (BEV) projection [44, 25, 63, 38, 5, 37] enables the recent TPVFormer [26] to generate surrounding 3D occupancy predictions by effectively exploiting tri-perspective view representations [13] augmenting the standard BEV with two additional perpendicular planes to recover the full 3D. All prior methods are trained in a supervised manner requiring rich voxel-level semantic information, which is costly to curate and annotate. While we build on [26], we forego manual label supervision and, instead, develop a model able to produce semantic 3D occupancy

predictions using supervision from LiDAR and from an image-language model allowing our model to acquire open-vocabulary skills in the voxel space.

**Multi-modal representation learning.** Distilling signals and knowledge from one modality into another is an effective strategy to learn representations [2, 3] or to learn to solve tasks using only few [14, 42, 1] or no human labels [54, 55]. The interplay between images, language and sounds is often used for self-supervised representation learning over large repositories of unlabeled data fetched from the internet [2–4, 41, 42]. Images can be paired with different modalities towards solving complex 2D tasks, e.g., semantic segmentation [55], detection of road objects [54] or sound-emitting objects [14, 42, 1]. Image-language aligned models project images and text into a shared representation space [21, 51, 34, 36, 19, 47, 28]. Contrastive image-language learning on many millions of image-text pairs [47, 28] leads to high-quality representations with impressive zero-shot skills from one modality to the other. We use CLIP [47] for its appealing open-vocabulary property that enables the querying of visual content with natural language toward recognizing objects of interest without manual labels. POP-3D uses LiDAR supervision for precise occupancy prediction and learns to produce in the 3D space CLIP-like features easily paired with language.

**Open-vocabulary semantic segmentation.** The aim of zero-shot semantic segmentation is to segment object classes that are not seen during training [59, 9, 24]. The advent of CLIP [47], which is trained on abundant web data, has inspired a new wave of methods, dubbed *open-vocabulary*, for recognizing random objects via natural language queries. CLIP features can be projected into 3D meshes [27] and NeRFs [29] to enable language queries. Originally producing image-level embeddings, CLIP can be extended to pixel-level predictions for open-vocabulary semantic segmentation by exploiting different forms of supervision from segmentation datasets, e.g., pixel-level labels [33] or class agnostic masks [20, 39, 62] coupled with region-word grounding [23], however with potential forgetting of originally learned concepts [27]. MaskCLIP+ [64] adjusts the attentive-pooling layer of CLIP to generate pixel-level CLIP features that are further distilled into an encoder-decoder semantic segmentation network. MaskCLIP+ [64] preserves the open-vocabulary properties of CLIP, and we exploit it here to distill its knowledge into POP-3D. We generate target 3D CLIP features by mapping MaskCLIP+ pixel-level features to LiDAR points observed in images. By being trained to match these distillation targets, POP-3D manages to learn 3D features with open-vocabulary perception abilities, in contrast to prior work on 3D occupancy prediction that is limited to recognizing a closed-set of visual concepts.

## 3 Open-vocabulary 3D occupancy prediction

Our goal is to predict 3D voxel representations of the environment, given a set of 2D input RGB images, that is amenable to open-vocabulary tasks such as zero-shot semantic segmentation or concept search driven by natural language queries. This is a challenging problem as we need to address the following two questions. First, what is the right architecture to handle the 2D-to-3D ambiguity and the open-vocabulary nature of the task? Second, how to formulate the learning problem without requiring manual annotation of large amounts of 3D voxel data, which are extremely hard to produce.

To address these questions we propose the following two innovations. First, we design an architecture for open-vocabulary 3D occupancy prediction (Fig. 2(a) and Sec. 3.1) that handles the 2D-to-3D prediction and open-vocabulary tasks with two specialized heads. Second, we formulate its training as a *tri-modal self-supervised learning* problem (Fig. 2(b) and Sec. 3.2) that leverages aligned (i) 2D images with (ii) 3D point clouds equipped with (iii) pre-trained language-image features as the three input modalities (i.e. camera, LiDAR and language) without the need for any explicit manual annotations. The details of these contributions are given next.

### 3.1 Architecture for open-vocabulary 3D occupancy prediction

We are given a set of surround-view images captured from one camera location and our goal is to output a 3D occupancy voxel map and to support language-driven tasks. To reach the goals, we propose an architecture composed of three modules (Fig. 2(a)). First, a *2D-3D encoder* predicts a voxel feature grid from the input images. Second, the *occupancy head* decodes this entire voxel grid into an occupancy map, predicting which voxels are free and which are occupied. Finally, the *3D-language head* is applied on each occupied voxel to output a powerful language embedding vector enabling a range of 3D open-vocabulary tasks. The three modules are described next.

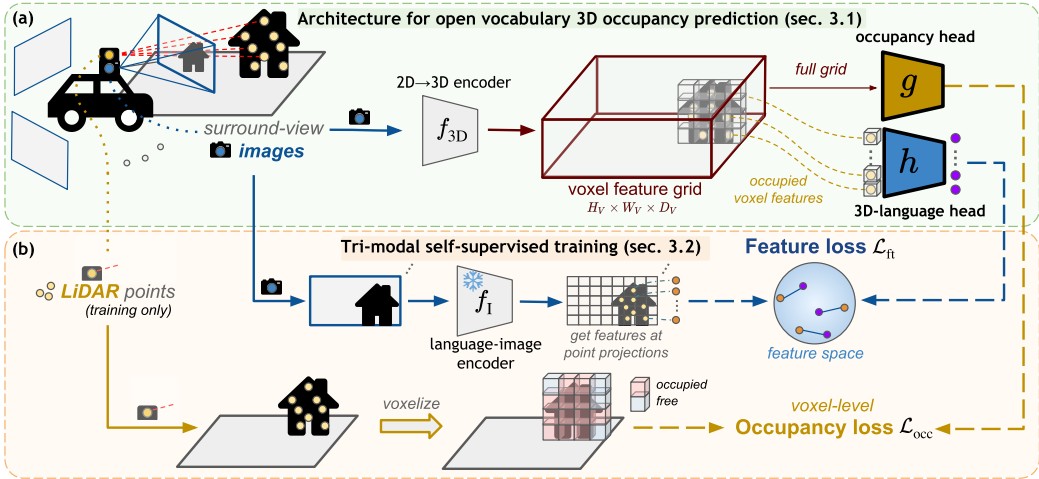

Figure 2: **Proposed approach.** In **(a)**, we show the architecture of the proposed method. Having only surround-view images on the input, the model first extracts dense voxel feature grid that is then fed to two parallel heads: occupancy head $g$ producing voxel-level occupancy predictions, and to 3D-language feature head $h$ which outputs features aligned with text representations. In **b)**, we show how we train our approach, namely the occupancy loss $\mathcal{L}_{\text{occ}}$ used to train class-agnostic occupancy predictions, and the feature loss $\mathcal{L}_{\text{ft}}$ that enforces the 3D-language head $h$ to output features aligned with text representations.

**2D-to-3D encoder $f_{\text{3D}}$.** The objective of the 2D-to-3D encoder is to predict a dense feature voxel grid given one or more images captured at one location as input. The output voxel grid representation encodes 3D visual information captured by the cameras. In detail, given surround-view camera RGB images $\mathbf{I}$ and camera calibration parameters, the encoder $f_{\text{3D}}$ produces a feature voxel grid

$$\mathbf{V} = f_{\text{3D}}\left(\mathbf{I}\right) \in \mathbb{R}^{H_{\text{V}} \times W_{\text{V}} \times D_{\text{V}} \times C_{\text{V}}}, \tag{1}$$

where $H_{\text{V}}, W_{\text{V}}$, and $D_{\text{V}}$ are the spatial dimensions of the voxel grid, and $C_{\text{V}}$ is the feature dimension of each voxel. This feature voxel grid is then passed to two distinct prediction heads designed to perform *class-agnostic occupancy prediction* and *text-aligned feature prediction* tasks respectively. The two heads are described next.

**Occupancy head $g$.** Given the feature voxel grid $\mathbf{V}$, the occupancy prediction head $g$ aims at classifying every voxel as 'empty' or 'occupied'. Following [26], this head is implemented as a non-linear network composed of $N_{\text{occ}}$ hidden blocks with configuration `Linear-Softplus-Linear`, each with $C_{\text{occ}}^{\text{hidden}}$ hidden features, and a final linear classifier outputting two logits, one per class. It outputs the tensor

$$\mathbf{O}_{\text{occ}} = g\left(\mathbf{V}\right) \in \mathbb{R}^{H_{\text{V}} \times W_{\text{V}} \times D_{\text{V}} \times 2}, \tag{2}$$

containing the occupancy prediction for each voxel.

**3D language head $h$.** In parallel, the voxel grid $\mathbf{V}$ is fed to a language feature extractor. This head processes each voxel feature to output an embedding vector that is aligned to vision-language representations, such as CLIP [47], aiming to inherit their open-vocabulary abilities. This allows us to address the limitations of closed-vocabulary predictions encountered in supervised 3D occupancy prediction models, which are bound to a set of predefined visual classes. In contrast, our representation enables us to perform 3D language-driven tasks such as zero-shot 3D semantic segmentation. Similarly to the occupancy head, the 3D-language head consists of $N_{\text{ft}}$ blocks with configuration `Linear-Softplus-Linear`, where each linear layer outputs $C_{\text{ft}}^{\text{hidden}}$ features, and a final linear layer that outputs $C_{\text{ft}}^{\text{out}}$-dimensional vision language embedding for each voxel. It outputs the tensor

$$\mathbf{O}_{\text{ft}} = h\left(\mathbf{V}\right) \in \mathbb{R}^{H_{\text{V}} \times W_{\text{V}} \times D_{\text{V}} \times C_{\text{ft}}^{\text{out}}}, \tag{3}$$

containing the predicted vision-language embedding of each voxel.

## 3.2 Tri-modal self-supervised training

The goal is to train the network architecture described in Sec. 3.1 to predict the 3D occupancy map together with language-aware features for each occupied voxel. In turn, this will enable 3D open-vocabulary tasks such as 3D zero-shot segmentation or language-driven search. The main challenge is obtaining the appropriate 3D-grounded language annotations, which is expensive to do manually. Instead, we propose a tri-modal self-supervised learning algorithm that leverages three modalities: (i) images, (ii) language and (iii) LiDAR point clouds. Specifically, we employ a pre-trained image-language network to generate image-language features for the input images. These features are then mapped to the 3D space using registered LiDAR point clouds, resulting in 3D grounded image-language features. These grounded features serve as training targets for the network. The training algorithm is illustrated in Fig. 2(b). The training is implemented via two losses that are used to train the two heads of the proposed architectures jointly with the 2D-to-3D encoder. The details are given next.

**Occupancy loss.** We guide the occupancy head $g$ to perform a class-agnostic occupancy prediction by the available unlabeled LiDAR point clouds, which we convert to occupancy prediction targets $T_{\text{occ}} \in \{0, 1\}$. Each voxel location $x$ containing at least one LiDAR point is labeled as 'occupied' (i.e., $T_{\text{occ}}(x) = 1$) and as 'empty' otherwise ($T_{\text{occ}}(x) = 0$). Having these targets, we supervise the occupancy prediction head densely at all locations of the voxel grid. The occupancy loss $\mathcal{L}_{\text{occ}}$ is a combination of cross-entropy loss $\mathcal{L}_{\text{CE}}$ and Lovász-softmax [6] loss $\mathcal{L}_{\text{Lov}}$:

$$\mathcal{L}_{\text{occ}}\left(\mathbf{O}_{\text{occ}}, \mathbf{T}_{\text{occ}}\right) = \mathcal{L}_{\text{CE}}\left(\mathbf{O}_{\text{occ}}, \mathbf{T}_{\text{occ}}\right) + \mathcal{L}_{\text{Lov}}\left(\mathbf{O}_{\text{occ}}, \mathbf{T}_{\text{occ}}\right), \tag{4}$$

where $\mathbf{O}_{\text{occ}}$ is the predicted occupancy tensor and $\mathbf{T}_{\text{occ}}$ the tensor of corresponding occupancy targets.

**Image-language distillation.** Unlike the occupancy prediction head that is supervised densely at the level of voxels, we supervise the 3D-language head at the level of points $p_n \in P_{\text{cam}}$ which project to at least one of the cameras, i.e., $P_{\text{cam}} \subset P$, where $P$ is the complete point cloud. This is required in order to obtain feature targets from the language-image pre-trained model $f_{\text{I}}$.

To get a feature target for a 3D point $p_n \in P_{\text{cam}}$ in the voxel feature grid, we use the known camera projection function $\Pi_c$ that projects 3D point $p_n$ into 2D point $u_n = (u_n^{(x)}, u_n^{(y)})$, where $(u_n^{(x)}, u_n^{(y)})$ are $(x, y)$ coordinates of point $u_n$ in camera $c$:

$$u_n = \Pi_c\left(p_n\right). \tag{5}$$

This way, we get a set of 2D points $U = \{\Pi_c\left(p_n\right)\}_{n=1}^{N}$ in the camera coordinates. To obtain feature targets $\mathbf{T}_{\text{ft}}$ for 3D points in $P_{\text{cam}}$ with corresponding 2D projections $U$ in camera $c$, we run the language-image-aligned feature extractor $f_{\text{I}}$ on image $\mathbf{I}_c$, and use the 2D projections' coordinates to sample from the resulting feature map, i.e.,

$$\mathbf{T}_{\text{ft}} = \left\{f_{\text{I}}\left(\mathbf{I}_c\right)\left[u_n^{(x)}, u_n^{(y)}\right]\right\}_{n=1}^{N} \in \mathbb{R}^{N \times C_{\text{ft}}^{\text{out}}}, \tag{6}$$

where $[x, y]$ is an indexing operator in the extracted feature map.

To train the 3D language head, we use $L_2$ mean squared error loss between the targets $\mathbf{T}_{\text{ft}}$ and the predicted features $\tilde{\mathbf{O}}_{\text{ft}} \in \mathbb{R}^{N \times C_{\text{ft}}^{\text{out}}}$ computed from $h$ for the 3D point locations in $P_{\text{cam}}$:

$$\mathcal{L}_{\text{ft}} = \frac{1}{N C_{\text{ft}}^{\text{out}}} \|\mathbf{T}_{\text{ft}} - \tilde{\mathbf{O}}_{\text{ft}}\|^2, \tag{7}$$

where $\| \cdot \|$ is the Frobenius norm.

**Final loss.** The final loss used to train the whole network is a weighted sum of the *occupancy* and *image-language* losses. We use a single hyperparameter $\lambda$ to balance the weighting of the two losses:

$$\mathcal{L} = \mathcal{L}_{\text{occ}} + \lambda \mathcal{L}_{\text{ft}}. \tag{8}$$

## 3.3 3D open-vocabulary test-time inference

Once trained, as described in Sec. 3.2, our model supports different 3D open-vocabulary tasks at test-time. We focus on the following two: (i) zero-shot 3D semantic segmentation and (ii) language-driven 3D grounding.

**Zero-shot 3D semantic segmentation from images.** Given an input test image, the 3D-text-aligned voxel features produced by our model support zero-shot 3D segmentation for a target set of classes specified via input text queries (prompts), as illustrated in Fig. 1. Unlike supervised approaches that necessitate retraining when the set of target classes changes, our approach requires training the model only once. We can adjust the number of segmented classes effortlessly by providing a different set of input text queries. In detail, at test-time we proceed along the following steps. First, a set of test surround-view images $\mathbf{I}$ from one location is fed into the trained POP–3D network, resulting in class-agnostic occupancy prediction $\mathbf{O}_{\text{occ}}$ via the occupancy head $g$, and language-aligned feature predictions $\mathbf{O}_{\text{ft}}$ via the 3D-language head $h$. Next, as described in [22], we generate a set of query sentences for each text query using predefined templates. These queries are then input into the pre-trained language-image encoder $f_{\text{text}}$, resulting in a set of language features. To obtain a single text feature per query, we compute the average of these features. Finally, considering $M$ such averaged text features, one for each of the $M$ target segmentation classes, we measure their similarity to the predicted language-aligned features $\mathbf{O}_{\text{ft}}$ at occupied voxels obtained from $\mathbf{O}_{\text{occ}}$. We assign the label with the highest similarity to each occupied voxel.

**Language-driven 3D grounding.** The task of language-driven 3D grounding is performed in a similar manner. However, here only a single input language query is given. Once determining the occupied voxels from $\mathbf{O}_{\text{occ}}$, we compute the similarity between the input text query encoded via the language-image encoder $f_{\text{text}}$ and predicted language-aligned features $\mathbf{O}_{\text{ft}}$ at the occupied voxels. The resulting similarity score can be visualized as a heat-map, as shown in Fig. 1, or thresholded to obtain the location of the target query.

# 4 Experiments

This section studies architecture design choices and demonstrates the capabilities of the proposed approach. First, in Sec. 4.1, we describe the experimental setup used, particularly the dataset, metrics, proposed evaluation protocol, and implementation details. Then, we compare our model to the state of the art in Sec. 4.2. Next, we present a set of studies on training hyperparameter sensitivity in Sec. 4.3 and finally show qualitative results in Sec. 4.4.

## 4.1 Experimental setup

We test the proposed approach on autonomous driving data, which provides a challenging test-bed.

**Dataset.** We use the nuScenes [10] dataset composed of 1000 sequences in total, divided into 700/150/150 scenes for train/val/test splits. Each sequence consists of $30 - 40$ scenes resulting in $28,130$ training and in $6,019$ validation scenes. The dataset provides 3D point clouds captured with 32-beam LiDAR, surround-view images obtained from six cameras mounted at the top of the car, and projection matrices between the 3D point cloud and cameras. LiDAR point clouds are annotated with 16 semantic labels. When using subsets of the complete dataset for ablations, we sort the scenes by their timestamp and take every $N$-th scene, e.g., every second scene in the case of a 50% subset.

**Metrics.** To evaluate our models on the task of 3D occupancy prediction, we need to convert the point-level semantic annotations from LiDAR to voxel-level annotations. We do this by taking the most-present label inside each voxel. As we aim at semantic segmentation, our main metric is mean Intersection over Union (mIoU), which we use in the evaluation protocol proposed in the next paragraph. Additionally, we measure the class-agnostic occupancy Intersection over Union (IoU). For the retrieval benchmark, we report the average precision (AP) for each query, the mean of which over all queries yields the mean average precision (mAP).

**New benchmark for open-vocabulary language-driven 3D retrieval.** To evaluate the retrieval capabilities, we collected a new *language-driven 3D grounding & retrieval benchmark equipped with natural language queries*. To build this benchmark, we annotated 3D scenes from various splits of the nuScenes dataset with the ground-truth spatial localization for a set of natural language open-vocabulary queries. The resulting set contains 105 samples in total, which are divided to 42/28/35 samples from train/val/test splits of the nuScenes dataset. The objective is, given the query, to retrieve all relevant 3D points from the LiDAR point cloud. Results are evaluated using the precision-recall curve; negative data are all the non-relevant 3D points in the given scene. For the evaluation purposes, we report numbers on a concatenated set consisting of samples from the validation and test splits

(63 samples). To annotate the 3D retrieval ground truth, we (1) manually provide the bounding box of the relevant object(s) in the image domain, (2) use Segment Anything Model [31] guided by our manual bounding box to produce a binary mask of this object, (3) project the LiDAR point cloud into the image, and (4) assign each 3D point a label corresponding to its projection into the binary mask. Furthermore, we use HDBSCAN [40] to filter points that are projected to the mask in the image but in fact do not belong to the object. This resolves the imprecisions caused by projection.

**New evaluation protocol for 3D occupancy prediction.** The relatively new task of 3D occupancy prediction has no established evaluation protocol yet. TPVFormer [26] did not introduce any evaluation protocol and provided only qualitative results. Having semantic labels only from LiDAR points, i.e., not in the target voxel space, makes it challenging to evaluate. Since voxel semantic segmentation consists of both *occupancy prediction* of the voxel grid and *classification of occupied voxels*, it is not enough to evaluate just at the points of ground-truth information from the LiDAR, as this does not take free space prediction into account.

To tackle this, we take inspiration from [7] and propose to obtain the evaluation labels from the available LiDAR point clouds, as depicted in Fig. 3 and described next. First, LiDAR rays passing through 3D space set the labels of intersected voxels to *free*. Second, voxels containing LiDAR points are assigned the most frequent semantic label of points lying within (or an *occupied label* in the case of class-agnostic evaluation). Third, all other voxels are *ignored* during evaluation, as they were not observed by any LiDAR ray and we are not certain whether they are occupied or not.

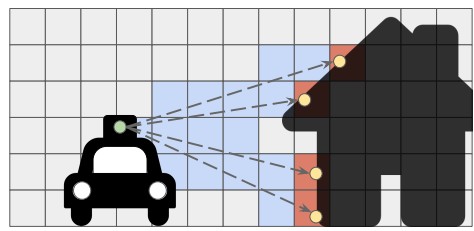

Figure 3: **Validation labels**: blue = *free*, red = *occupied*, and gray = *ignored* voxels.

**Implementation details.** We use the recent TPVFormer [26] as backbone for the 2D-3D encoder. It takes surround-view images on the input and produces a voxel grid of size $100 \times 100 \times 8$, which corresponds to the volume $[-51.2m, +51.2m] \times [-51.2m, +51.2m] \times [-5m, +3m]$ around the car. For the language-image feature extractor, we use MaskCLIP + [64], which provides features of dimension $C_{\text{ft}}^{\text{out}} = 512$. If not mentioned otherwise, we use the default learning rate of 2e-4, Adam [30] optimizer, and a cosine learning rate scheduler with final learning rate 1e-6, and with linear warmup from 1e-5 learning rate for the first 500 iterations. We train our models on 8×A100 GPUs. We use ResNet-101 as image backbone in the $f_{3D}$ encoder, and full-scale images on the input. Both prediction heads have two layers, i.e., $N_{\text{occ}} = N_{\text{ft}} = 2$, and $C_{\text{occ}} = 512$ and $C_{\text{ft}} = 1024$ feature channels. With this architecture setup, we train our model on 100% of the nuScenes training data for 12 epochs. We put the same weight to the occupancy and feature losses, i.e., we set $\lambda = 1$ in Eq. 8. We ablate these choices in Sec. 4.3.

## 4.2 Comparison to the state of the art

Here we compare our approach to four relevant methods: (i) the fully supervised (closed-vocabulary) TPVFormer [26] and the following three open-vocabulary image-based methods, namely to (ii) MaskCLIP+ [64], (iii) ODISE [60], and (iv) OpenScene [43], which require 3D LiDAR point clouds on the input during the inference. Please note that compared to methods (ii)-(iv), our POP-3D does not require (1) strong manual annotations (either in the image or point cloud domain) or (2) having point clouds on the input during the inference. Details are given next.

**Comparison to a fully-supervised TPVFormer [26].** In figure Fig. 4b, we compare our results to the supervised TPVFormer [26] in terms of class-agnostic IoU and (16+1)-class mIoU (16 semantic classes plus the *empty* class) on the nuScenes [10] validation set. Interestingly, our model outperforms its supervised counterpart in the class-agnostic IoU by 11.5 points, showing superiority in the prediction of the occupied space. This can be attributed to different training schemes of the two methods: in the fully-supervised case, the *empty* class competes with the other semantic classes, whereas in our case the occupancy head performs only class-agnostic occupancy prediction. Next, for the (16+1)-class semantic occupancy segmentation, we can see that our zero-shot approach reaches ≈ 78% of the supervised counterpart performance, which we consider as strong result given that the latter requires manually annotated point clouds for training. In contrast, our approach is zero-shot and does not require any manual point cloud annotations at training. These results pave the way

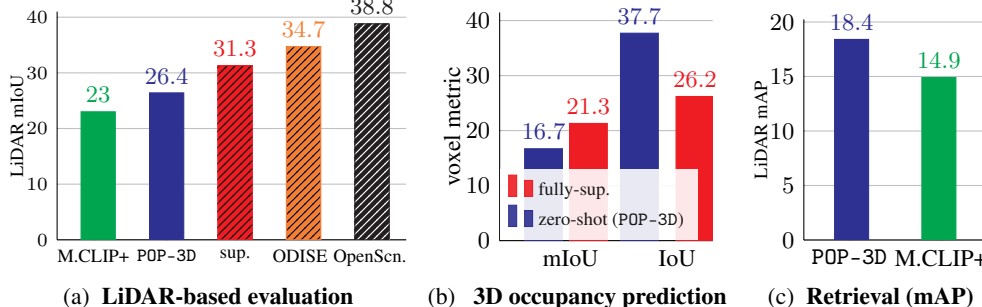

(a) **LiDAR-based evaluation**  (b) **3D occupancy prediction**  (c) **Retrieval (mAP)**

Figure 4: **Comparison to the state of the art**. We compare our POP-3D approach to different baselines using (a) the LiDAR-based evaluation, (b) occupancy evaluation, and (c) open-vocabulary language-driven retrieval. In (a), our zero-shot approach POP-3D outperforms the strong MaskCLIP+ [64] (M.CLIP+) baseline, while closing the gap to the fully supervised. Other recent methods using supervision and requiring LiDAR points during inference (ODISE [60] and Open-Scene [43]) are even better. All methods that require manual annotations during training are denoted by striped bars). In (b), our zero-shot approach POP-3D surpasses the fully-supervised model [26] on occupancy prediction (IoU) while reaching 78% of its performance on semantic occupancy segmentation (mIoU). Finally, in (c) we present results of open-vocabulary language-driven retrieval on our newly composed dataset, where we compare our approach to the MaskCLIP+ baseline. We measure mAP on manually annotated LiDAR 3D points in the scene. Our POP-3D outperforms the MaskCLIP+ approach on this task by 3.5 mAP points.

for language-driven vision-only 3D occupance prediction and semantic segmentation in automotive applications. We show qualitative results of our POP-3D approach in Fig. 5 and in the supplementary materials.

**Comparison to MaskCLIP+ [64].** In Fig. 4a we compare the quality of the 3D vision-language features learnt by our POP-3D approach against the strong *MaskCLIP+*[64] baseline. In detail, we project the 3D LiDAR points to the 2D image(s) space, sample MaskCLIP+[64] features extracted from the 2D image at the projected locations and backproject those extracted features back to 3D via the LiDAR rays. Note that *MaskCLIP+* features are used in our tri-modal training to represent the language modality so it is interesting to evaluate the benefits of our approach in comparison to directly transferring *MaskCLIP+* features to 3D. For a fair comparison, we evaluate only the LiDAR points with a projection to the camera, i.e., this evaluation considers only the classification of the 3D points, not the occupancy prediction itself. We call this metric LiDAR mIoU. Our POP-3D outperforms MaskCLIP+ (26.4 vs. 23.0 mIoU), i.e., our method learns better 3D vision-language features than its teacher, while also not requiring LiDAR data at test time (as MaskCLIP+ does). Finally, Fig. 4a shows that POP-3D reaches ≈ 84% of the performance of the fully-supervised model [26].

**Comparison to open-vocabulary methods that require additional supervision.** Furthermore, we compare our approach to ODISE [60] and OpenScene [43], which both require manual supervision during training. ODISE requires panoptic segmentation annotations for training, while OpenScene uses features from either LSeg [33] or OpenSeg [20], which are two image-language encoders that are trained with supervision from manually provided segmentation masks. We report results using OpenSeg. As Fig. 4b shows, these methods perform best, which can be attributed to additional manual annotations available during training.

**Open-vocabulary language-driven retrieval.** The goal is, given a text query of the searched object, to retrieve all 3D points belonging to the object in the given scene. During the evaluation, to get the relevance of LiDAR points to the query text description, we follow the same approach as for the task of zero-shot semantic segmentation, i.e., we pass the images to our model, get features aligned with the text, and compute their relevance to the given text query. This gives a score for every 3D point in the scene. In the ideal case the points belonging to the target object should have the highest score. We compare our method with MaskCLIP+ and report results in Fig. 4c. Our approach exhibits superior mAP compared to MaskCLIP+, achieving 18.4 mAP while MaskCLIP+ obtains mAP of 14.9.

## 4.3 Sensitivity analysis

Here we study the sensitivity of our model to various hyperparameters. Except otherwise stated, for this study we use half-resolution input images, i.e., $450 \times 800$, the ResNet-50 backbone, and train for 6 epochs using 50% of the nuScenes training data.

Table 1: **Sensitivity analysis.** We investigate here the impact of loss weight $\lambda$ in the final loss function (a), the image resolution and image backbone (b) and the depth of the prediction heads (c).

(a) **Loss weight $\lambda$ impact**

| $\lambda$ | mIoU | IoU |
|---|---|---|
| 1.00 | 12.0 | 30.0 |
| 0.50 | 12.0 | 30.5 |
| 0.25 | 11.9 | 30.5 |

(b) **Image resolution and backbone**

| image resolution | mIoU | |
|---|---|---|
| | RN50 | RN101 |
| $450 \times 800$ | 12.0 | 15.1 |
| $900 \times 1600$ | 12.3 | 15.2 |

(c) **Depth of prediction heads**

| $N_{occ}$ / $N_{ft}$ | mIoU | |
|---|---|---|
| | 2 | 3 |
| 2 | 15.4 | 15.3 |
| 3 | 15.3 | 15.5 |

**Loss weight $\lambda$.** In Tab. 1a we study the sensitivity of our model to the loss weight $\lambda$ of $\mathcal{L}_{ft}$ in Eq. 8. We see that the model's performance is not sensitive to $\lambda$. By default we use $\lambda = 1$.

**Input resolution and image backbone.** In Tab. 1b we experiment with (a) using half ($450 \times 800$) or full ($900 \times 1600$) input images, and (b) using ResNet-50 (RN50) or ResNet-101 (RN101) for the image backbone. Following [26], RN50 is initialized from MoCov2 [15] weights and RN101 from FCOS3D [56] weights. We see that it is better to use the RN101 backbone while the input resolution has small impact (with full resolution being better).

**Depth of prediction head.** In Tab. 1c we study the impact of the $N_{occ}$ and $N_{ft}$ hyperparameters that control the number of hidden layers on the occupancy prediction head $g$ and 3D language head $h$ respectively, using RN101 as backbone. We see that the depth of the two prediction heads does not play a major role and it is slightly better to be the same, i.e, $N_{occ} = N_{ft}$. Therefore, we opt to use $N_{occ} = N_{ft} = 2$ in our experiments, as it performs well and requires less compute.

## 4.4 Demonstration of open-vocabulary capabilities

In Fig. 6 we provide visualizations of language-based 3D object retrievals inside a scene using text queries like "building door" and "tire". For reference, green boxes denote locations of reference objects (cars), to ease the orientation in the scene. The results show that our model is able to localize in 3D space fine-grained language queries.

**Limitations.** First, given the low spatial resolution of the voxel grid our model does not discover well small objects. This is not a limitation of the method, but of the currently used backbone architecture and input data. Second, another limitation is that our architecture does not natively support sequences of images as input which might be beneficial for reasoning about semantic occupancy of occluded objects and areas appearing thanks to relative motion of objects in the scene.

## 5 Conclusion

In this paper we propose POP-3D, a tri-modal self-supervised learning strategy with a novel architecture that enables open-vocabulary voxel segmentation from 2D images and at the same time improves the occupancy grid estimation by a significant margin over the state of the art. Our approach also outperforms the strong baseline of directly back-projecting 2D vision-language features into 3D via LiDAR and does not require LiDAR at test-time. This work opens-up the possibility of large-scale open-vocabulary 3D scene understanding driven by natural language.

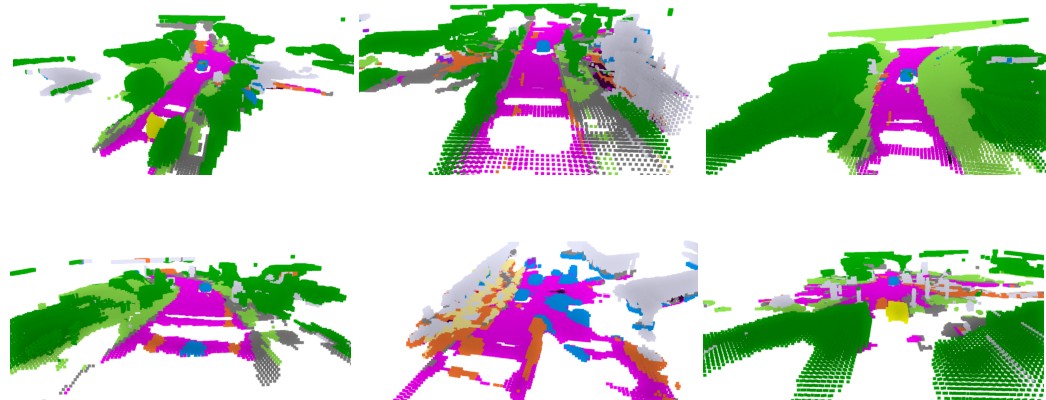

Figure 5: **Qualitative results of zero-shot semantic 3D occupancy prediction** on the 16 classes in the nuScenes [10] validation split. Please note how our method is able to quite accurately localize and segment objects in 3D including road (magenta), vegetation (dark green), cars (blue), or buildings (gray) from only input 2D images and in a zero-shot manner, i.e. only by providing natural language prompts for the target classes. Visualizations are shown on an interpolated 300x300x24 voxel grid.

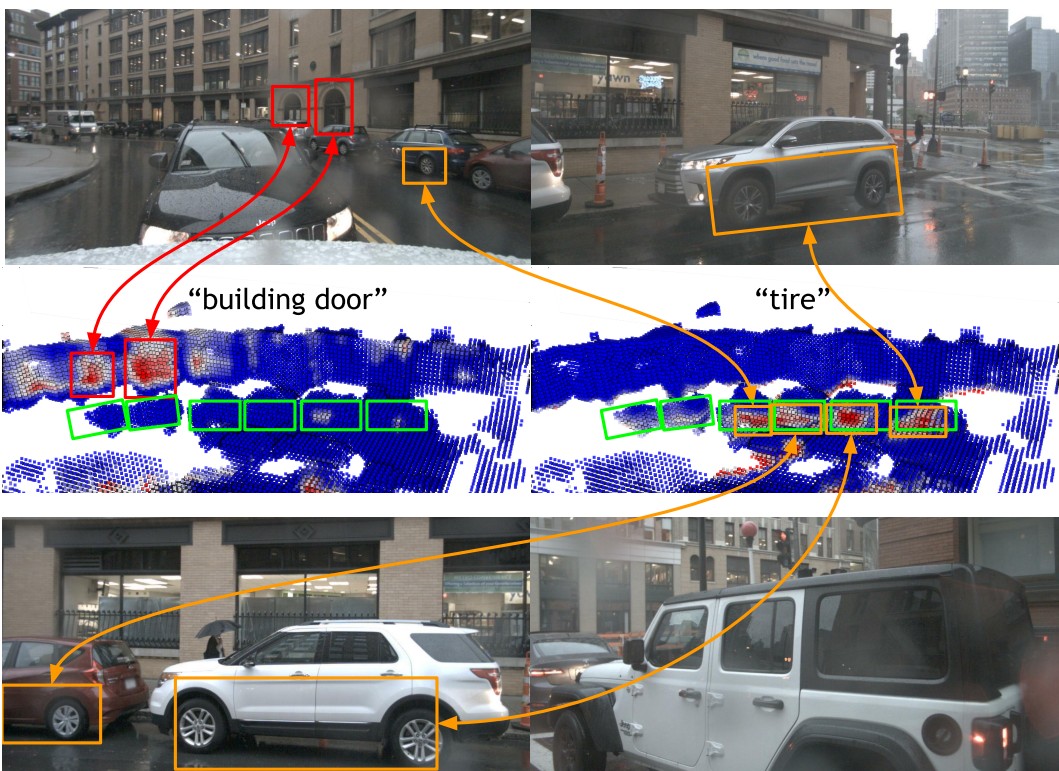

Figure 6: **Qualitative results showcasing the language-driven 3D grounding and retrieval.** On left (in red) we can see the six input images passed to the POP-3Dto get open-vocabulary 3D features (middle). Given the searched object name ("Black hatchback"), we compute the similarity with the 3D feature field and obtain similarity heatmap (right). Language-based 3D retrieval is not possible using existing close-vocabulary methods such as [26]. Please see additional results in the supplementary.

## Acknowledgements

This work supported by the European Regional Development Fund under the project IMPACT (no. CZ.02.1.010.00.015_0030000468), by the Ministry of Education, Youth and Sports of the Czech Republic through the e-INFRA CZ (ID:90140), and by CTU Student Grant SGS21184OHK33T37. This research received the support of EXA4MIND, a European Union's Horizon Europe Research and Innovation programme under grant agreement N° 101092944. Views and opinions expressed are however those of the author(s) only and do not necessarily reflect those of the European Union or the European Commission. Neither the European Union nor the granting authority can be held responsible for them. The authors have no competing interests to declare that are relevant to the content of this article. Antonin Vobecky acknowledges travel support from ELISE (GA no 951847). We acknowledge the support from Valeo.

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
