# POP-3D: Open-Vocabulary 3D Occupancy Prediction from Images
# – Supplementary Material –

**Antonin Vobecky**[1,2,3]     **Oriane Siméoni**[1]     **David Hurych**[1]     **Spyros Gidaris**[1]

**Andrei Bursuc**[1]          **Patrick Pérez**[1]          **Josef Sivic**[2]

[1] valeo.ai, Paris, France    [2] CIIRC CTU in Prague    [3] FEE CTU in Prague
https://vobecant.github.io/POP3D

In this supplementary material, we first give additional details about the method in Sec. 1. Then, in Sec. 2, we provide additional details about the benchmark for open-vocabulary language-driven 3D retrieval. Finally, we present additional qualitative results in Sec. 3.

## 1   Text queries for zero-shot 3D occupancy prediction

In this section, we investigate how selecting text queries assigned to specific ground-truth classes impacts semantic segmentation.

To simplify the analysis of the impact of the language prompt, we study MaskCLIP+ [3] features, which we also use as our training targets. This choice allows us to uncover the capabilities associated with these features. Using the nuScenes [1] dataset, we project the language-image-aligned features from MaskCLIP+ onto the corresponding LiDAR points. To measure the mIoU, we evaluate our approach on a subset comprising 25% of the nuScenes validation set. It is important to note that, for a fair comparison, we calculate the mIoU only for the points with camera projections (other LiDAR points cannot have associated features from MaskCLIP+).

**Queries used for zero-shot semantic segmentation.**    To get the text queries for the task of language-guided zero-shot semantic segmentation, we utilize the textual descriptions from the nuScenes [1] dataset associated with every sub-class (names of the sub-classes are in the first column of Tab. 1). We parse these descriptions into a set of queries (for every sub-class) and show them in the last column in Tab. 1. We do this for all the annotated classes in the dataset (second column).

**Limited-classes experiment.**    First, we conduct a controlled experiment with a limited set of five classes that are described by names 'car,' 'drivable surface,' 'pedestrian,' 'vegetation,' and 'manmade' in the nuScenes [1] dataset. We refer to this specific setup as `original-5`, disregarding the other classes for the purpose of this study. One can see that, for example, class name *'manmade'* lacks descriptive specificity. In the text description of this class, we can find "... buildings, walls, guard rails, fences, poles, street signs, traffic lights ..." and more. We make similar observations for a number of class names in the nuScenes [1] dataset. This observation highlights the limitation of relying solely on class names to guide text-based querying.

To study and address this limitation, we introduce two additional setups, namely `manmade-5` and `pedestrian-5`. In `manmade-5`, we replace the class name *'manmade'* with *'building'*, while in

---

[2]Czech Institute of Informatics, Robotics and Cybernetics at the Czech Technical University in Prague

Table 1: **Queries used for zero-shot semantic segmentation.** We take the textual descriptions from the nuScenes [1] dataset associated with every sub-class (names of the sub-classes are in the first column) and use them to get the individual queries associated with this class (the last column). We do this for all the annotated classes in the dataset.

| Sub-class | Training class | Derived descriptions/queries |
| --- | --- | --- |
| noise | noise | "any lidar return that does not correspond to a physical object, such as dust, vapor, noise, fog, raindrops, smoke and reflections" |
| adult pedestrian | pedestrian | adult |
| child pedestrian | pedestrian | child |
| construction worker | pedestrian | construction worker |
| personal mobility | ignore | skateboard; segway |
| police officer | pedestrian | police officer |
| stroller | ignore | stroller |
| wheelchair | ignore | wheelchair |
| barrier | barrier | "temporary road barrier to redirect traffic"; concrete barrier; metal barrier; water barrier |
| debris | ignore | "movable object that is left on the driveable surface"; tree branch; full trash bag |
| pushable pullable | ignore | "object that a pedestrian may push or pull"; dolley; wheel barrow; garbage-bin; shopping cart |
| traffic cone | traffic cone | traffic cone |
| bicycle rack | ignore | "area or device intended to park or secure the bicycles in a row" |
| bicycle | bicycle | bicycle |
| bendy bus | bus | bendy bus |
| rigid bus | bus | rigid bus |
| car | car | "vehicle designed primarily for personal use"; car; vehicle; sedan; hatch-back; wagon; van; mini-van; SUV; jeep |
| construction vehicle | construction vehicle | vehicle designed for construction.; crane |
| ambulance vehicle | ignore | ambulance; ambulance vehicle |
| police vehicle | ignore | police vehicle; police car; police bicycle; police motorcycle |
| motorcycle | motorcycle | motorcycle; vespa; scooter |
| trailer | trailer | trailer; truck trailer; car trailer; bike trailer |
| truck | truck | "vehicle primarily designed to haul cargo"; pickup; lorry; truck; semi-tractor |
| driveable surface | driveable surface | "paved surface that a car can drive"; "unpaved surface that a car can drive" |
| other flat | other flat | traffic island; delimiter; rail track; stairs; lake; river |
| sidewalk | sidewalk | sidewalk; pedestrian walkway; bike path |
| terrain | terrain | grass; rolling hill; soil; sand; gravel |
| manmade | manmade | man-made structure; building; wall; guard rail; fence; pole; drainage; hydrant; flag; banner; street sign; electric circuit box; traffic light; parking meter; stairs |
| other static | ignore | "points in the background that are not distinguishable, or objects that do not match any of the above labels" |
| vegetation | vegetation | bushes; bush; plants; plant; potted plant; tree; trees |
| ego vehicle | ignore | "the vehicle on which the cameras, radar and lidar are mounted, that is sometimes visible at the bottom of the image" |

pedestrian-5, we use *'person'* instead of *'pedestrian'*. The results presented in the upper part of Tab. 2 demonstrate the effectiveness of these changes. Specifically, replacing *'manmade'* with *'building'* improves the IoU for this category from 17.4 to 45.1, and using *'person'* instead of *'pedestrian'* increases the IoU from 1.3 to 14.6 for the respective class. These findings highlight the suboptimal use of original class names as text queries.

**Training-classes experiment.** Building upon these findings, we extend our study to include the full set of 16 classes used in the nuScenes dataset. We conduct experiments using two setups: i) original-16, which uses the original training class names from the nuScenes dataset, and ii)

Table 2: **Segmentation mIoU with a different number of target classes and text queries**. The first part of the table considers segmentation into 5 classes only, while the second part evaluates the complete set of 16 training classes. Results were obtained using 25% of the validation split.

| setup | visible points IoU | | | | | |
|---|---|---|---|---|---|---|
| {NAME}-{#cls} | mIoU | car | road | ped. | veg. | man. |
| **5 classes** | | | | | | |
| original-5 | 27.5 | 21.2 | 37.3 | 1.3 | 60.3 | 17.4 |
| manmade-5 | **34.7** | 28.1 | 37.3 | 1.6 | 61.2 | **45.1** |
| pedestrian-5 | **35.0** | 17.5 | 61.9 | **14.6** | 59.7 | 21.3 |
| **16 classes** | | | | | | |
| original-16 | 10.2 | 25.8 | 0.9 | 3.0 | 51.3 | 0.5 |
| descriptions-16 | **23.0** | 37.9 | 57.5 | 16.9 | 62.6 | 45.4 |

`descriptions-16`, where we utilize the detailed textual descriptions that are provided for each class in the nuScenes dataset (we explain in more details this setup in the next paragraph). By leveraging the textual descriptions provided by the nuScenes dataset, we can generate more informative and descriptive queries for each individual class, as demonstrated in Tab. 1. This table presents the entire set of 32 (sub-)classes annotated in the nuScenes [1] dataset, along with their mapping to the training classes and the corresponding derived queries. The lower section of Tab. 2 demonstrates the impact of modifying the text queries associated with individual classes in the nuScenes dataset. We observe that this simple adjustment significantly enhances the mIoU from 10.2 to 23.0, highlighting the significance of query selection. Based on these results, we have used the `descriptions-16` setup for our experiments in the main paper.

The obtained results suggest that further improvements could be achieved by tuning the text queries carefully. However, it is important to note that the focus of our paper is not on exploring query tuning, and therefore, we do not delve further into this direction.

**Using derived descriptions for segmentation.** To utilize the derived queries presented in Tab. 1, we begin by mapping the 32 sub-classes to the 16 training classes in the nuScenes dataset (note that some sub-classes are marked as 'ignore' in the "Training class" column of Tab. 1 to indicate that they are actually ignored during evaluation). As an example, let's consider the training class '*pedestrian*'. The sub-classes that are associated to this training class are: '*adult pedestrian*', '*child pedestrian*', '*construction worker*' and '*police officer*'. We use the derived text descriptions (third column in Tab. 1) of each of those sub-classes as text queries for the '*pedestrian*' training class, resulting in the following set of queries: [*adult, child, construction worker, police officer*].

This process produces a set of queries $Q$ with size $N^Q = |Q| = \sum_{c \in \{0...15\}} N_c^Q$, where $N_c^Q$ is number of queries associated with the training class $c \in \{0 \ldots 15\}$. Each query $q \in Q$ is mapped to a single training class $c$ via the mapping:

$$\mathcal{M} : \{0 \ldots N^Q - 1\} \rightarrow \{0 \ldots 15\}. \tag{1}$$

To assign a feature $o_{\mathrm{ft}}$ from the set of all predicted features $O_{\mathrm{ft}}$ to one of the training classes, we follow a three-step process. First, we calculate the similarity between the feature $o_{\mathrm{ft}}$ and each query. Next, we select the query with the highest similarity. Finally, we assign the corresponding training class label $c_{\mathrm{pred}}$ based on the selected query. For example, if the query '*police officer*' has the highest similarity, we assign the label '*pedestrian*' to the feature $o_{\mathrm{ft}}$. This can be formulated as:

$$c_{\mathrm{pred}} = \mathcal{M} \left( \arg\max_{n \in \{0...N^Q - 1\}} (\mathrm{sim} (o_{\mathrm{ft}}, q_n)) \right), \tag{2}$$

where $q_n$ is the $n$-th query.

## 2 Benchmark for open-vocabulary language-driven 3D retrieval

In Tab. 3, we present queries contained in the retrieval benchmark together with the number of queries in individual splits.

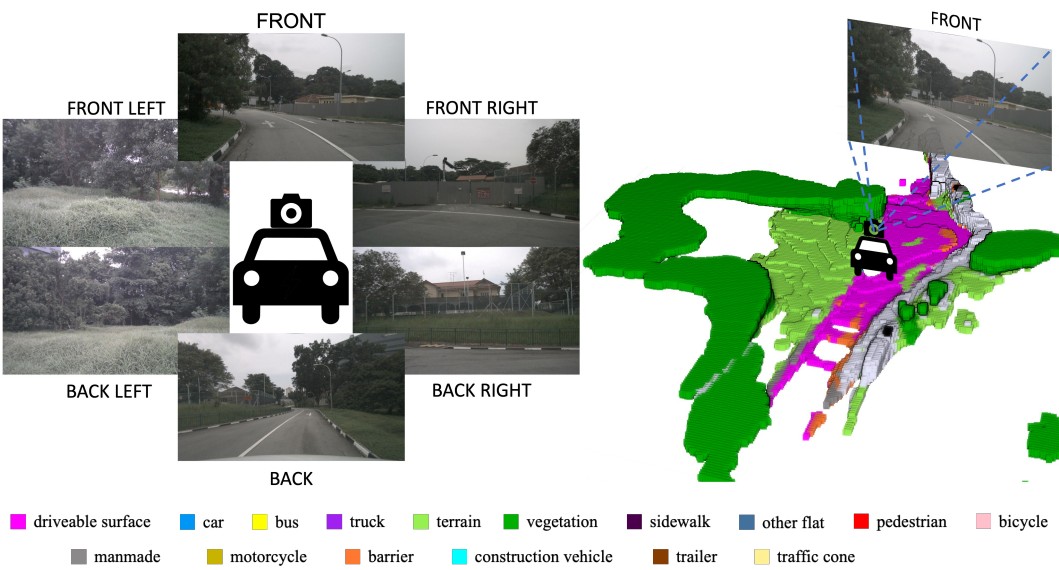

FRONT

FRONT LEFT

FRONT RIGHT

BACK LEFT

BACK RIGHT

BACK

FRONT

| | | | | | |
|---|---|---|---|---|---|
| ■ driveable surface | ■ car | ■ bus | ■ truck | ■ terrain | ■ vegetation |
| ■ sidewalk | ■ other flat | ■ pedestrian | ■ bicycle | | |
| ■ manmade | ■ motorcycle | ■ barrier | ■ construction vehicle | ■ trailer | ■ traffic cone |

Figure 1: **Zero-shot 3D occupancy prediction**. *Left:* six input surround-view images. *Right:* our prediction; training grid resolution 100×100×8 is upsampled to 300×300×24 by interpolating the trained representation space.

## 3 Qualitative results

In this section, we first show additional qualitative results for the task of zero-shot 3D occupancy prediction using 16 classes in the nuScenes [1] dataset in Figures 1, 2 and 4. We further proceed with qualitative examples of the retrieval task in Figures 5 and 6.

**Zero-shot 3D occupancy prediction.** In Figures 1, 2, and 4, we present qualitative results of zero-shot 3D occupancy prediction for 16 semantic categories in the nuScenes dataset [1]. These figures showcase the ability of our method to reconstruct the overall 3D structure of the scene accurately. Moreover, as shown in Figure 2, our method can recognize classes such as *bus*, which are not well represented in the training dataset.

**Text-based retrieval.** We present qualitative results of the retrieval task in Figures 5 and 6. These figures demonstrate the effectiveness of our model in retrieving non-annotated categories, such as *stairs* or *zebra crossing*, by querying the predicted features with a single text query and visualizing the similarity overlaid on the voxel grid. However, we observed limitations in the retrieval capabilities due to two factors: a) the resolution of the voxel grid and b) the level of concept granularity captured in MaskCLIP+ features. It is important to note that although MaskCLIP+ features (extracted with a DeepLabv2 architecture) have better spatial precision, they do not fully preserve all the descriptive capabilities of the original CLIP, as pointed out also in [2], since they are already distilled from the CLIP model.

**Qualitative comparison.** In Fig. 3, we present a qualitative comparison of our POP-3D to fully-supervised TPVFormer and to MaskCLIP+ results projected from 2D to 3D ground-truth point cloud.

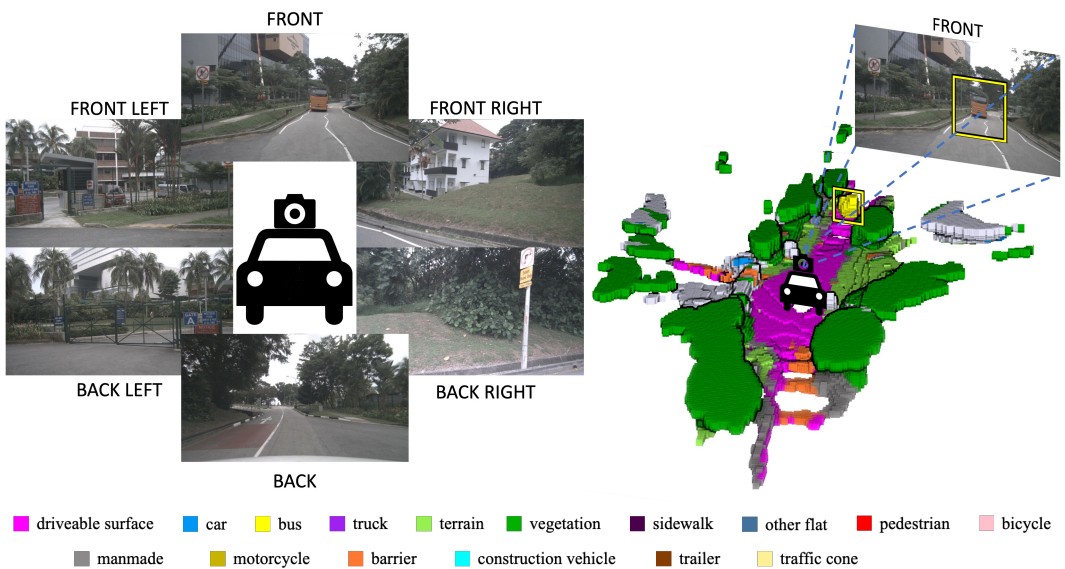

| ■ driveable surface | ■ car | ■ bus | ■ truck | ■ terrain | ■ vegetation | ■ sidewalk | ■ other flat | ■ pedestrian | ■ bicycle |
|---|---|---|---|---|---|---|---|---|---|
| ■ manmade | ■ motorcycle | ■ barrier | ■ construction vehicle | ■ trailer | ■ traffic cone | | | | |

Figure 2: **Zero-shot 3D occupancy prediction**. *Left:* six input surround-view images. *Right:* our prediction; training grid resolution 100×100×8 is upsampled to 300×300×24 by interpolating the trained representation space. It is worth noting that the model successfully segments even the class *bus*, despite its limited occurrence in the training set.

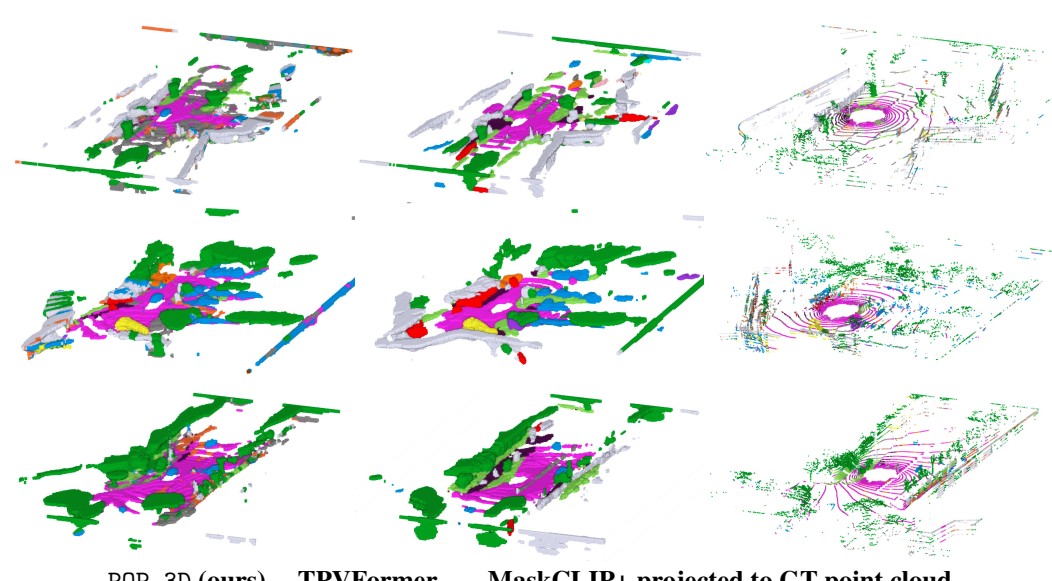

POP-3D **(ours)**     **TPVFormer**     **MaskCLIP+ projected to GT point cloud**

Figure 3: **Qualitative results.** Qualitative comparison of our POP-3D to fully-supervised TPVFormer and to MaskCLIP+ results projected from 2D to 3D ground-truth point cloud.

Table 3: **Number of queries in the individual splits of the retrieval benchmark dataset**.

| query | val | test | train | |
|---|---|---|---|---|
| agitator truck | 0 | 0 | 1 | 1 |
| bulldozer | 0 | 1 | 1 | 2 |
| excavator | 3 | 1 | 1 | 5 |
| asphalt roller | 0 | 0 | 1 | 1 |
| dustcart | 1 | 0 | 0 | 1 |
| boom lift vehicle | 0 | 1 | 4 | 5 |
| sedan | 1 | 0 | 0 | 1 |
| sports car | 1 | 0 | 0 | 1 |
| hatchback | 1 | 0 | 0 | 1 |
| mini-van | 0 | 0 | 1 | 1 |
| van | 0 | 0 | 1 | 1 |
| lorry | 0 | 0 | 2 | 2 |
| wagon | 0 | 0 | 1 | 1 |
| SUV | 1 | 1 | 0 | 2 |
| jeep | 1 | 1 | 0 | 2 |
| campervan | 0 | 0 | 1 | 1 |
| motorcycle | 0 | 0 | 1 | 1 |
| vespa with driver | 0 | 0 | 1 | 1 |
| golf cart | 0 | 1 | 0 | 1 |
| forklift | 0 | 0 | 1 | 1 |
| scooter with rider | 0 | 0 | 1 | 1 |
| skateboard with rider | 0 | 1 | 0 | 1 |
| ice cream van | 0 | 1 | 0 | 1 |
| parcel delivery vehicle | 1 | 1 | 0 | 2 |
| food truck | 0 | 0 | 1 | 1 |
| police car | 0 | 1 | 0 | 1 |
| police van | 0 | 0 | 1 | 1 |
| dog | 0 | 0 | 1 | 1 |
| bird | 0 | 0 | 2 | 2 |
| double decker bus | 1 | 0 | 1 | 2 |
| pick up truck for human transport | 0 | 0 | 1 | 1 |
| jogger | 0 | 0 | 1 | 1 |
| stroller | 0 | 0 | 1 | 1 |
| two persons walking together | 1 | 0 | 0 | 1 |
| person with a leaf blower | 0 | 0 | 1 | 1 |
| chair | 0 | 1 | 0 | 1 |
| stairs | 1 | 0 | 1 | 2 |
| horse sculpture | 1 | 0 | 0 | 1 |
| vase | 0 | 0 | 1 | 1 |
| traffic lights | 0 | 0 | 1 | 1 |
| fire hydrant | 1 | 2 | 1 | 4 |
| mailbox | 3 | 0 | 0 | 3 |
| mailboxes | 0 | 0 | 1 | 1 |
| suitcase | 0 | 0 | 1 | 1 |
| wheelbarrow | 0 | 0 | 1 | 1 |
| garbage bin | 0 | 0 | 1 | 1 |
| cardboard box | 0 | 0 | 1 | 1 |
| mirror | 0 | 1 | 0 | 1 |
| human with an umbrella | 1 | 0 | 2 | 3 |
| rain barrel | 0 | 0 | 1 | 1 |
| mobile toilet | 1 | 0 | 0 | 1 |
| pedestrian crossing | 1 | 0 | 0 | 1 |
| barrier gate | 0 | 0 | 2 | 2 |
| motorbike | 0 | 1 | 0 | 1 |
| yellow school bus | 0 | 1 | 0 | 1 |
| police officer | 0 | 2 | 0 | 2 |
| chopper | 0 | 1 | 0 | 1 |
| small bulldozer | 0 | 1 | 0 | 1 |
| concrete mixer truck | 0 | 2 | 0 | 2 |
| truck crane | 0 | 1 | 0 | 1 |
| cabriolet | 0 | 1 | 0 | 1 |
| yellow car | 0 | 1 | 0 | 1 |
| baby stroller | 0 | 2 | 0 | 2 |
| green trash bin | 0 | 1 | 0 | 1 |
| red sedan | 1 | 1 | 0 | 2 |
| delivery van | 0 | 1 | 0 | 1 |
| white truck tractor | 0 | 1 | 0 | 1 |
| keg barrels | 0 | 1 | 0 | 1 |
| regular-cab truck | 0 | 1 | 0 | 1 |
| minibus | 0 | 1 | 0 | 1 |
| trash bin | 1 | 1 | 0 | 2 |
| white suv | 1 | 0 | 0 | 1 |
| delivery truck | 1 | 0 | 0 | 1 |
| black truck with a trailer | 1 | 0 | 0 | 1 |
| person on a bicycle | 1 | 0 | 0 | 1 |
| scooter | 1 | 0 | 0 | 1 |
| total | 28 | 35 | 42 | 105 |

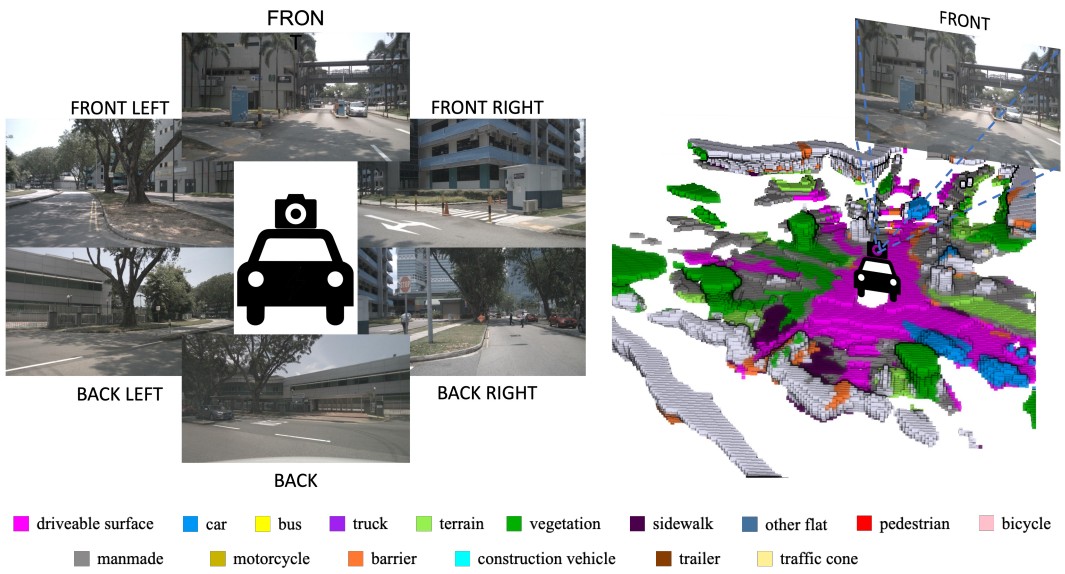

Figure 4: **Zero-shot 3D occupancy prediction**. *Left:* six input surround-view images. *Right:* our prediction; training grid resolution $100\times100\times8$ is upsampled to $300\times300\times24$ by interpolating the trained representation space. This example demonstrates the model's ability to accurately reconstruct complex 3D scenes.

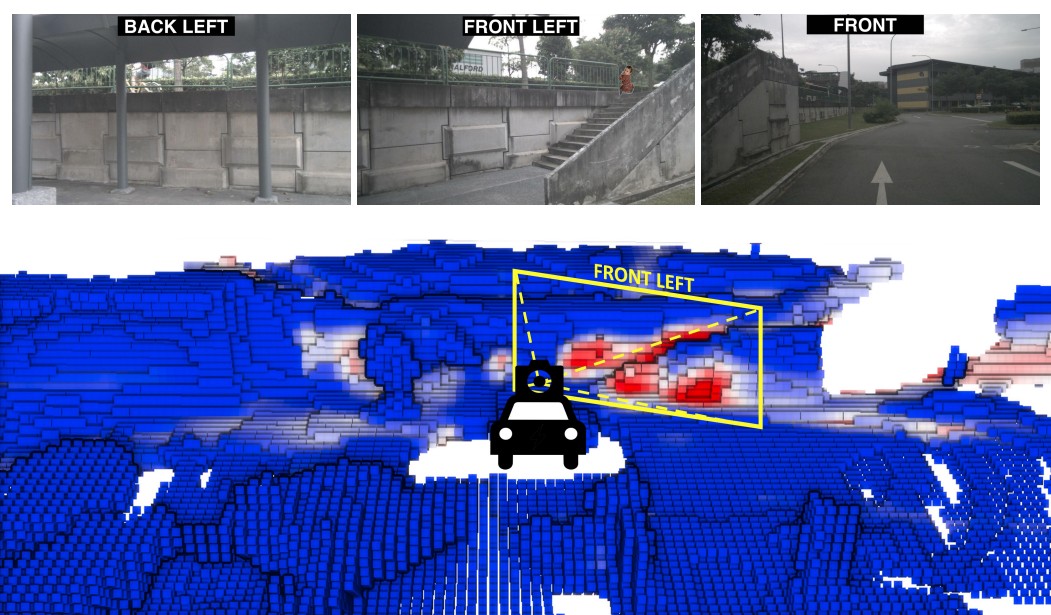

Figure 5: **Text-based 3D retrieval.** Query: '*stairs*'. Top: Input images showing just the left side of the car. Bottom: a detailed view of the corresponding 3D scene with a heatmap indicating the similarity to the '*stairs*' query. The stairs are mostly visible in the front left camera, as also highlighted in the predicted 3D scene.

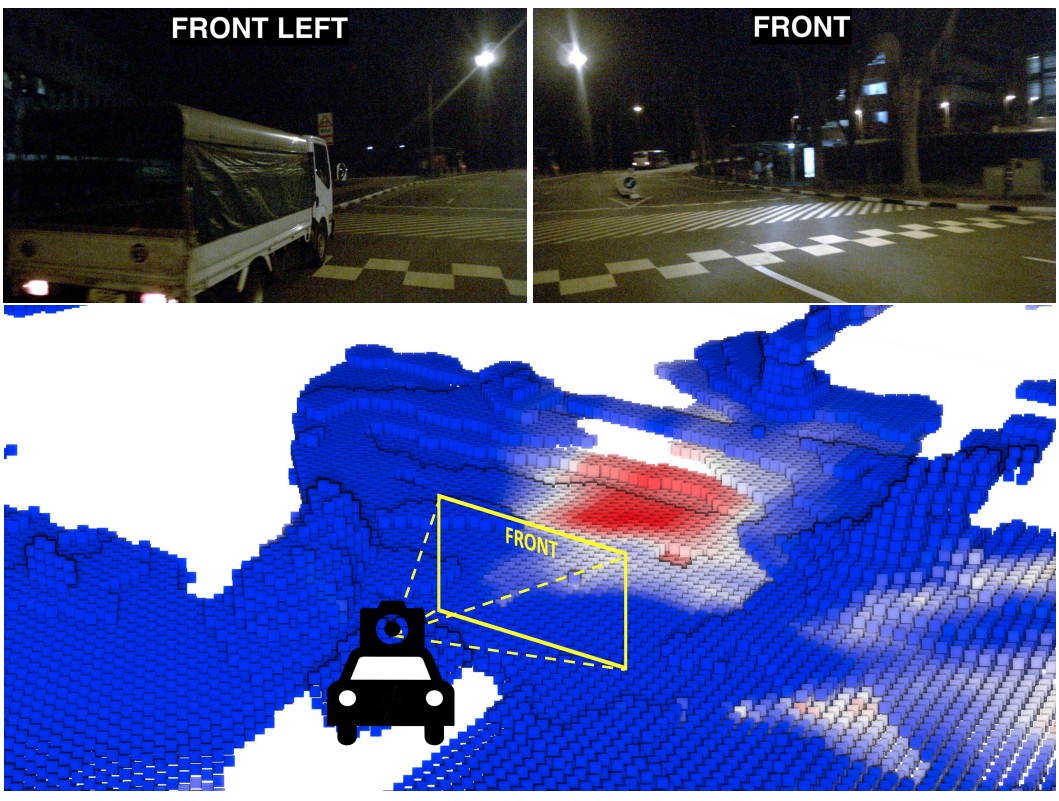

Figure 6: **Text-based 3D retrieval.** Query: '*zebra crossing*'. Top: Input images showing the scene in front of the car. Bottom: a detailed view of the corresponding 3D scene with a heatmap indicating the similarity to the '*zebra crossing*' query. This example showcases the model's ability to recognize fine-grained concepts, even under challenging conditions like nighttime.