# OpenReview forum: "POP-3D: Open-Vocabulary 3D Occupancy Prediction from Images"
_NeurIPS.cc/2023/Conference — NeurIPS 2023 poster_

### Official Review · Reviewer_q7fa · 2023-07-03

**Soundness:** 2 fair
**Presentation:** 3 good
**Contribution:** 2 fair
**Rating:** 5
**Confidence:** 5

**Summary:**

This paper proposes an approach, POP-3D, to achieving open-vocabulary 3D occupancy prediction from images by aligning 3D features with those 2D features from pre-trained MaskCLIP+. Specifically, in contrast to previous occupancy prediction methods, such as TPVFormer, this framework uses two heads to predict class-agnostic occupancy and semantic-related features, respectively and then uses the features to perform language-grounded open-vocabulary 3D perception. To evaluate the effectiveness of the method, the paper also introduces a new evaluation method tailored to the 3D occupancy prediction task. Promising experimental results show the efficacy of the proposed method.

**Strengths:**

- The basic idea is easy to follow and the illustration figures are clear.
- The motivation to address the problems of open-vocabulary 3D occupancy prediction is good and the proposed method is effective.
- The design of decomposed heads for geometry and semantic prediction is reasonable.
- The proposed self-supervised learning method does not need 3D manual annotations while achieving comparable performance with supervised methods.
- The experiments compared with MaskCLIP+ baseline are interesting, giving a more comprehensive position of the proposed method among different solutions for the 3D occupancy prediction task.

**Weaknesses:**

- (Method) Although the proposed method can achieve open-vocabulary 3D perception, the basic capability is mainly borrowed from 2D features and the pre-trained MaskCLIP+. It is one of the reasonable solutions for open-vocabulary 3D perception but I have to say there is nothing new for exploring how to achieve that from 3D data. I can understand that this is also related to the task setting in this paper is 3D occupancy prediction from "images", but it makes the studied problem more like a "pseudo-3D open-vocabulary" one.

- (Problem Setting) While this paper shows the open-vocabulary capability of the proposed method with the zero-shot occupancy prediction results and the visualization of the case study, it still does not provide a clearly useful background to perform 3D open-vocabulary occupancy prediction in "driving scenarios" (there are typically very few categories of interest, which are almost enough for safe planning and control), or at least provide a good playground/annotations to study this problem. This significantly weakens the value of this paper, because the foundation of studying this problem is not very clear, especially when this benchmark is not officially set up or popular enough such that we do not need to justify this.

- (Evaluation) The introduced evaluation protocol mainly considers the weaknesses of "sparse" semantic occupancy prediction in TPVFormer and adds empty labels along a lidar ray. However, such problems do not exist in many recent occupancy benchmarks [1][2][3], where dense occupancy labels are available. Therefore, I think the contribution of the new evaluation protocol is a little incremental and does not solve the mentioned problem fundamentally compared to these new occupancy benchmarks.

[1] Tian et. al., Occ3D: A Large-Scale 3D Occupancy Prediction Benchmark for Autonomous Driving

[2] Tong et. al., Scene as Occupancy

[3] Wei et. al., SurroundOcc: Multi-Camera 3D Occupancy Prediction for Autonomous Driving

**Questions:**

None

**Limitations:**

The authors have mentioned the potential limitations and provide simple clarifications and possible solutions.

---

> ### Author Rebuttal · Authors · 2023-08-09
>
> We thank the reviewer for their feedback.
>
> **Method novelty**
>
> We would like to thank the reviewer for the comment. Although it is true that POP-3D uses image-text aligned features provided by MaskCLIP+, we argue that exploiting such features is an interesting and non-trivial starting point to explore the hard task of Open Vocabulary 3D semantic occupancy.
>
> There are indeed several recent works that distill CLIP features into another modality, e.g., audio, point clouds, etc., to inherit CLIP’s open-vocabulary capabilities. While POP-3D exploits information from LiDAR to understand the 3D geometry of a scene, the supervisory signal comes from the image domain where large-scale language-image pre-training datasets (such as those used for training CLIP) are available, it remains an image-based method. We argue that the novelty and difficulty of our endeavor is in reaching an effective interplay between the complementary information coming from multiple camera images, LiDAR geometry, and language-image features in the complex outdoor 3D space surrounding the vehicle, without using any manual labels. The quantitative results showing performance boosts in occupancy estimation (IoU - Figure 4b) and close performance on semantic predictions (mIoU - Figure 4b), while outperforming original MaskCLIP+ features (Figure 4a), confirm that POP-3D does not just imitate image-language features, but reaches non-trivial 3D perception capabilities.
>
> **Motivation of open vocabulary in driving scenarios**
>
> The need for open vocabulary in driving scenarios is essential for statistical system validation. In contrast to the standard research dataset setup (majority of data for training, minority for validation and testing), testing, called system validation, of the Advanced Driving Assistance Systems (ADAS) requires vastly more data captures than training. As an example, a level 3 system validation requires tens of thousands of hours of real-world driving in order to satisfy requirements based on functional safety standards defined in ISO 26262. Open-vocabulary functionality is essential to understand the contents of the data for: (a) being able to find out if specific long tail distribution object classes are present in the data based on system requirements and (b) having the ability to recognize the root causes of failures (e.g., specific object types on which false positives appear), in order to have feedback to models training and adaptation to improve performance and increase safety. In short, the behavior of ADAS on objects from classes unseen at training time needs to be thoroughly tested for safety reasons, and such object classes are unknown in advance.
>
> We argue that there are many non-frequent types of objects in driving scenarios that are essential for the autonomous driving perception system to work under all conditions. To support this, we follow the suggestion of the reviewer and prepare a small benchmark to study this problem. We refer the reviewer to the section “Language-driven 3D grounding & retrieval evaluation” in the General Response. So far, we gathered seven scenes with non-frequent objects, such as excavators, large trash bin, or police car. We present the results in Table A of the General Response. We aim to enlarge this benchmark and publish it alongside the final paper. We hope that this can serve as a playground to study this problem.
>
> **Limitation of evaluation protocol and new occupancy benchmarks**
>
> We agree with the comments of the reviewer regarding the limitations of our evaluation protocol. We thank the reviewer for the pointers to the recent and elaborate benchmarks with dense labels that are relevant to our work. However, we would like to highlight that the suggested benchmarks were published on arXiv in March 2023 (SurroundOcc [3]), April 2023 (Occ3D [1]), and June 2023 (OccNet [2]), so below the two-month period suggested by NeurIPS guidelines.
>
> Due to differences in the granularity of the voxel size produced by POP-3D and the multiple ones from the benchmarks we were unable to adapt and evaluate our method for such an experiment during the short rebuttal time. However, we plan to improve our evaluation protocol by leveraging these new findings and conduct evaluations on them for the updated version of the paper.

---

> > ### Comment · Reviewer_q7fa · 2023-08-15
> > **Response to Rebuttal**
> >
> > I acknowledge that I have read the authors' rebuttal and the other reviews.
> >
> > Thank you for addressing my concerns. I will keep my score and give borderline acceptance in the final decision. As the other reviewers may have important concerns not fully addressed, I would not argue my case if other reviewers reach a different recommendation.

---

### Official Review · Reviewer_Vrgc · 2023-07-04

**Soundness:** 3 good
**Presentation:** 4 excellent
**Contribution:** 2 fair
**Rating:** 6
**Confidence:** 4

**Summary:**

This work presents a system that is able to achieve open-vocabulary 3D semantic volumetric predictions with multi-view image inputs. Given a set of images, the system learns a 3D voxel grids which decodes into binary occupancy and visual-linguistic embeddings. The embeddings are used for retrieval and semantic segmentation given language inputs. Comparisons are present against the fully-supervised TPVFormer and  MaskCLIP+ on nuScenes.

**Strengths:**

- The paper is well-written, easy to read, and materials are clearly presented.
- The presented system should be quite useful in downstream applications especially for outdoor driving scenarios.
- The presented method is reasonable, well designed, and is new.

**Weaknesses:**

- The paper only compares to MaskCLIP+ as the zero-shot baseline (TPVFormer is fully supervised), which is insufficient, given the abundant related works like Lerf/Semantic-abs/OpenScene: 3D Scene Understanding with Open Vocabularies /CLIP-FO3D: Learning Free Open-world 3d Scene Representations from 2D Dense CLIP. Although these work may use NeRF/point clouds as the 3D representation, the reviewer finds the core technical challenges are shared.
- Given these previous works, which are not even cited and discussed, I would consider the technocal novelity of this work is low.
- No qualitative comparison to maskclip+/TVPFormer. Please show figures.


Some minor questions
- what's the design for f_3d? No explanation given for this image->3D lifting backbone.
- In line 162, how about some points are occupied but not observed in any view?

**Questions:**

see weankess

**Limitations:**

no issue found

---

> ### Author Rebuttal · Authors · 2023-08-09
>
> We thank the reviewer for the feedback.
>
> **Comparison to other open-vocabulary methods: LeRF, Semantic-Abs, OpenScene, CLIP-FO3D**
>
> Thank you for suggesting relevant works which tackle a similar problem to ours. We will integrate a discussion about those works in our related work section.
>
> For now, we would like to point out important differences between each one of them and our POP-3D method. The first, LeRF, is a NeRF-based method, which needs to be trained on each scene independently and does not generalize to novel scenes. CLIP-FO3D is a 3D network processing point clouds and RGB images and learns to imitate MaskCLIP features via pairs of RGB images aligned with dense point clouds as input during inference, whereas we produce 3D open-vocabulary semantic occupancy maps only from images. OpenScene learns a representation of 3D point clouds and is therefore limited to the availability of 3D scanners at inference time, which is too costly to be mounted on every car. Moreover, the CLIP-like features in OpenScene are obtained using either LSeg [A] (which requires ground-truth semantic image annotations) or OpenSeg [B] methods (which requires class-agnostic segmentation masks and image captions) which both need much more annotation than MaskCLIP+ (which requires no additional annotation) exploited in our POP-3D. Therefore, it is expected that LSeg and OpenSeg features provide better pixel-text alignment. In Table B of the General Response, we provide results using the OpenScene method (adapted to our evaluation framework and without test-time augmentations). OpenScene achieves better results than our method.  Again, we would like to stress that OpenScenes uses image-language encoders that were trained with much more supervision than MaskCLIP+, which we use in our experiments. At the same time, our method is agnostic to the used image-language encoder and could hence exploit encoders from LSeg and OpenSeg (as OpenScene does) to improve its performance.
>
> We would be interested to have the full reference to the "Semantic-abs" work mentioned by the reviewer, as we were not able to find it.
>
> **Technical novelty**
>
> We would like to point out the tight/impossible timeline here, which is the reason why we were not aware/couldn't compare to works suggested by the reviewer. Although we agree those works are relevant, two were published on arXiv in March 2023 (CLIP-FO3D, LERF) (which is below the two months suggested by NeurIPS guidelines), and OpenScene's (presented at CVPR23, which happened in June 2023) code was provided only in mid-March 2023.
>
> Nonetheless, as mentioned above, we thank the reviewer for pointing us out to those works and will discuss them in the related work section of the paper.
>
> **Qualitative results for MaskCLIP+ and TPVFormer**
>
> We provide an additional qualitative comparison of our POP-3D framework to the fully-supervised TPVFormer and to MaskCLIP+ results projected from 2D to 3D ground-truth point cloud. These qualitative results can be found in the PDF within the General Response.
>
> **Design of f_3d image-to-3D lifting**
>
> We use the TPVFormer [26] model, which builds upon the popular BEVFormer bird's-eye-view (BEV) lifting method [37]. BEVFormer is an attention-based lifting method using points from the BEV grid as queries for fetching visual information from the image encoder features of the different cameras. TPVFormer generalizes BEVFormer to the 3D space in a computationally efficient manner via a tri-perspective view (TPV) representation. Three axis-aligned orthogonal TPV planes (HW, DH, WD, with H,W,D denoting the resolution of the three planes across height, width, and depth dimensions) are learned with BEVFormer lifting. Voxels are modeled in the 3D space by summing their projected features on the three planes. We give information about the setup of the 3D backbone in the "Implementation details" subsection and will expand the provided description. Additional information can be found in the original TPVFormer paper.
>
> We would like to highlight that POP-3D is not specifically designed for use with TPVFormer backbone and could be used with other image-to-3D lifting strategies as long as they produce voxel-level representations, which is the case for all recent methods.
>
> **Occupied invisible points**
>
> Following the original implementation of TPVFormer, we set such voxels as empty. We are aware of potentially different solutions for setting such voxels as ignored, but we did not yet test this training setup.

---

> > ### Comment · Reviewer_Vrgc · 2023-08-22
> >
> > Thanks for your reply. I've read the rebuttal and the other reviews. I'd like to raise my score to Weak Accept.

---

### Official Review · Reviewer_9pAV · 2023-07-04

**Soundness:** 3 good
**Presentation:** 2 fair
**Contribution:** 2 fair
**Rating:** 5
**Confidence:** 5

**Summary:**

The paper presents a novel approach for predicting open-vocabulary 3D semantic voxel occupancy maps from 2D images. The objective is to enable 3D grounding, segmentation, and retrieval of free-form language queries, which is challenging due to the ambiguity between 2D and 3D representations and the difficulty of obtaining annotated 3D training data. The proposed model architecture consists of a 2D-3D encoder, occupancy prediction, and 3D-language heads. It generates a dense voxel map of 3D grounded language embeddings, facilitating various open-vocabulary tasks. The authors also introduce a tri-modal self-supervised learning algorithm that leverages images, language, and LiDAR point clouds to train the model without the need for manual 3D language annotations.

**Strengths:**

## Strengths
- Tri-Modal Self-Supervised Learning: This approach enables training the model using a pre-trained vision-language model without the need for any 3D manual language annotations.

- Experiment results: The authors quantitatively evaluated the model's performance in zero-shot 3D semantic segmentation using existing datasets. Additionally, the model's effectiveness in tasks such as 3D grounding and retrieval of free-form language queries is demonstrated.


**Weaknesses:**

## Weakness:
- The experiment is unable to demonstrate the advantage of the so-called “Open”-vocabulary. Only a few categories are demonstrated in the experiment. If we only need to understand a few categories, it would be better to just use standard object detection methods that have tens of common categories. At least hundreds of object categories need to be demonstrated to justify the “Open”, which is a key advantage that the authors claimed.
- The method needs paired lidar-image data. This makes it less generalizable than the masked clip.
- Although the proposed method is better than masked clip in some known categories, this might not be true for a wider range of categories unseen in training.


**Questions:**

Could the authors justify the paper's contribution given the weakness?

**Limitations:**

Please see weaknesses.

---

> ### Author Rebuttal · Authors · 2023-08-09
>
> We thank the reviewer for their feedback and suggestions. We address here each comment/question.
>
> **Evaluating open-vocabulary capabilities on hundreds of object categories**
>
> We would like to highlight that open-vocabulary approaches for image-based 2D tasks (semantic segmentation, object detection) can show results on a high number of classes also due to the existence of several richly annotated datasets with numerous classes (COCO, LVIS, Objects365), whereas most autonomous driving datasets come with a small number of annotated classes. We emphasize that for the 16 classes we consider in our evaluation, no class information was used during the self-supervised training, making this evaluation open-vocabulary. In contrast, similar approaches in the image domain achieve wider open-vocabulary detection skills by leveraging different forms of annotations, typically for a set of base classes (LSEG [A], Detic [E], OWL-ViT [F], etc.) and are then evaluated on a set of novel classes. POP-3D does not use any human annotation and learns to predict 3D semantic voxel occupancy maps only from LiDAR information and distillation from image-language features.
>
> We understand the reviewer is asking for the “open” aspect of the method to be more consistently evaluated. We were not able to gather a benchmark with hundreds of object categories in the one week of the rebuttal (no such dataset exists among automotive driving datasets with paired images and LiDAR scans). However, we produce a first small benchmark which we plan to expand. We refer the reviewer to the section "Language-driven 3D grounding & retrieval evaluation" in the General Response. We propose a new benchmark that aims to evaluate the "open" quality of methods. Although limited in terms of the number of queries, we observe that this preliminary benchmark demonstrates further the good performances of our method and better ones than MaskCLIP+.
>
> **Neccesity of paired LiDAR-image data and generalization compared to MaskCLIP**
>
> We would like to clarify this point. Indeed, the reviewer is correct in saying that our method POP-3D is trained using LIDAR-image pairs. However, during inference, our method takes **only 2D images as input** (no LiDAR data is used as input) and produces a 3D semantic open vocabulary feature map as output. In contrast, MaskCLIP does not need any paired LiDAR-image data for training but needs such pairs at inference time for the re-projection of features from 2D to 3D space. The MaskCLIP+ baseline is rather an artificial baseline that we proposed to emphasize that POP-3D successfully learns not only to lift and predict image-language features in the 3D space but also to leverage its 3D perception skills in order to produce better predictions. We observe that POP-3D outperforms MaskCLIP+, which needs LiDAR at inference time. We would like to stress that the setup of POP-3D reflects current ways of capturing datasets for autonomous driving [G,H,I,J] and also industry standards, where it is common to have paired image-LiDAR data during the initial data capture in order to validate the perception algorithms (such as depth prediction from camera validated by ground-truth measurements from LiDAR scanner). Such data can be captured using only a small fleet of cars. On the other hand, it is expensive to have LiDAR scanners mounted on every deployed car due to the high price of the sensor.
>
> **Performance on categories unseen in training**
>
> We would like to clarify that during training, POP-3D does not receive any explicit information about the “known” categories; the model “sees” the pixels of the objects in the images, their corresponding LiDAR points, and MaskCLIP+ features, but no labels. These categories are not known by the model and are given to it only at inference time in the form of text embeddings. We try to answer the lack of comparison in the long-tail classes by collecting a new benchmark for open-vocabulary language-driven 3D grounding and retrieval. We refer the reviewer to the corresponding section in the General Response. Table A there provides results with our method and MaskCLIP+. Our method achieves better retrieval performance.
>
> Regarding object categories that are not seen at all during training, i.e., not present in the images of the nuScenes dataset, we hypothesize that given the performance boosts over the MaskCLIP+ baseline, POP-3D learns to exploit both the structure of the CLIP feature space and the 3D word layout. As a result, it can deal to some extent with such categories depending on how far they are from the nuScenes data distribution.
>
> [E] Zhou et al., Detecting Twenty-thousand Classes using Image-level Supervision, ECCV 2022
>
> [F] Minderer et al., Simple Open-Vocabulary Object Detection with Vision Transformers, ECCV 2022
>
> [G] Sun et al., Scalability in perception for autonomous driving: Waymo open dataset, CVPR 2020
>
> [H] Geiger et al,. Are we ready for autonomous driving? the KITTI vision benchmark suite, CVPR 2012
>
> [I] Behley et al., SemanticKITTI: A dataset for semantic scene understanding of lidar sequences, ICCV 2019
>
> [J] Mao et al., One million scenes for autonomous driving: ONCE dataset, NeurIPS Datasets and Benchmarks 2021

---

> > ### Comment · Reviewer_9pAV · 2023-08-16
> > **Response to the Authors**
> >
> > I appreciate the authors' response and clarification. This paper has explored a new task where the spatial information from lidar and the semantic information from CLIP is distilled into the pure image-based network. Although there are some problems regarding the usefulness and practicality of the new task, I would still encourage this kind of work and its new exploration. I have raised my score.

---

### Official Review · Reviewer_yGBj · 2023-07-06

**Soundness:** 3 good
**Presentation:** 3 good
**Contribution:** 3 good
**Rating:** 5
**Confidence:** 4

**Summary:**

This paper proposes a camera-only open-vocabulary 3D occupancy prediction method, named POP3D. POP3D consists of a 2D-3D encoder and 3D language heads, which can combine open-vocabulary segmentation with 3D occupancy prediction together. The proposed method can be trained in a self-supervised manner by leveraging images, language, and Lidar modalities. Experiments on the widely used nuScenes benchmark demonstrate the effectiveness of the proposed method.

**Strengths:**

[1] The presentation of this paper is clear.
[2] The motivation of this paper makes sense.
[3] The proposed method has practical value in auto-labeling system.


**Weaknesses:**

The paper is well-writen overall. One of my concerns is that the evaluation protocol proposed by the authors may not accurately reflect the quality of occupancy prediction. Would you please provide more details about the groud truth generation? (e.g. How many lidar sweeps have you used? Because the Lidar used in nuScenes dataset is sparse than others.) I am concerned that inaccurate evaluations may lead to incorrect conclusions.

**Questions:**

Please see weaknesses.

**Limitations:**

Please see weaknesses.

---

> ### Author Rebuttal · Authors · 2023-08-09
>
> We thank the reviewer for the feedback. We address here the concerns individually.
>
> **Details about ground-truth generation**
>
> We happily provide more details about the ground-truth generation. We use a single LiDAR sweep (i.e., only a single LiDAR point cloud from a single time step without point cloud aggregation). Then we consider the volume  $[−51.2m,+51.2m]×[−51.2m,+51.2m]×[−5m,+3m]$ around the car and voxelize the space with cubic voxels of 1m size (see Sec. 4.1/Implementation details). Following [7], we cast rays from the LiDAR sensor mounted on the car to the points in the LiDAR point cloud (see Figure 3 in the paper for visualization). We use the same assumption as [7], i.e., we consider all the points along the ray as an empty space. We proceed with setting all the voxels that contain at least one point from the LiDAR point cloud as “occupied”. The remaining voxels are ignored, as we cannot be sure whether they are empty or occupied.
>
> We understand the concern of the reviewer about the sparsity of the nuScenes LiDAR, which has only 32 beams, hence a low vertical density. However, we argue that this does not lead to inaccurate occupancy predictions because the resolution in the horizontal direction is sufficient and allows the sensor to capture all objects except those extremely thin.

---

### Official Review · Reviewer_nph2 · 2023-07-07

**Soundness:** 3 good
**Presentation:** 3 good
**Contribution:** 3 good
**Rating:** 6
**Confidence:** 4

**Summary:**

This work presents a novel approach for zero-shot 3D occupancy prediction for autonomous driving applications. The key idea consists of three parts:
- a 2D-3D encoder to create a 3D voxel feature grid, based on TPVFormer
- An MLP-based voxel occupancy predictor that is class agnostic
- an MLP-based feature encoder, called a "3D language head" that takes the TPV-former features that have positive occupancy, and learns to predict the corresponding features of an off-the-shelf MaskCLIP+ model. In effect this is distilling the 2D MaskCLIP+ features into 3D.

At inference time LiDAR information is not necessary because occupancy is predicted directly from RGB images. Further, the model can be used zero-shot at inference time by computing the similarty of the output of the 3D language head features with language features from a CLIP text encoder. During training, only direct supervision is needed derived from class-agnostic occupancy from the LiDAR data, the distillation loss only needs an off-the-shelf vision language model. The proposed approach obtains ~78% of the fully supervised TPVFormer, while needing no class label supervision.

**Strengths:**

Originality
- While there are a variety of works trying to leverage vision-language models, this is the first work to show promising results on open vocabulary 3D occupancy prediction. The proposed model is simple but highly effective taking advantage of existing LIDAR datasets, the TPVFormer architecture, and off the shelf vision language models. The simplicity of this technique opens up the potential for removing the dependence of expensive supervision in this problem domain and instead focusing on extracting information from existing vision-language models.

Quality
- The main claim of the paper is that the proposed system can perform zero-shot semantic occupancy prediction without using any 3D semantic labels at training time. This paper provides sufficient evidence for this claim by evaluating on nuScenes and obtaining ~78% of  the performance of the fully supervised model. This highlights the power of vision-language models.
- The related work is detailed and positions the proposed work well relative to previous techniques semantic 3D occupancy prediction, multi-modal learning and open vocabulary segmentation.
- Section 4.3 presents empirical evidence for the choice of hyperparameters, and the effect of image resolutions.

Clarity
- The paper writing is excellent, it easy to follow and understand. Figure 2 is particularly helpful for following the technical description of the model in Section 3.

**Weaknesses:**

Quality - small scale evaluation
- The evaluation is done only on one dataset. This is understandable given the computation costs for training and evaluating models, but having evaluation on more than one dataset would significantly improve the paper. A potential candidate dataset is https://pandaset.org/ which contains LIDAR and 5 wide angle cameras.
- Evaluation is done only against one open-vocabulary baseline and one supervised basline. While TPVFormer appears to be the only fully supervised reasonable choice, there are baselines like ODISE (https://github.com/NVlabs/ODISE) or OVSeg (https://jeff-liangf.github.io/projects/ovseg/) for open vocabulary segmentation, whose predictions could get back-projected into the class-agnostic occupancy predictions, to obtain 3D semantic labels. These are relevant baselines in light of the comparison with MaskCLIP.
- Open-vocabulary recognition is powerful because it allows tackling difficult long-tailed scenarios. Currently such a benchmark doesn't exist for occupancy prediction, limiting the evidence to qualitative examples, if we want to evaluate performance beyond the classes defined in nuScenes. It would be a great improvement to the paper if there were some annotated samples to set quantitative benchmark.

**Questions:**

- Why is evaluation limited to only one dataset when there are other datasets that can be used for evaluation?
- Why aren't existing open-vocabulary segmentation models, backprojected in 3D, used as baselines in addition to MaskCLIP? This is particularly important given the small scale experiments.

**Limitations:**

The authors have adequately addressed the limitations and the potential negative societal impact.

---

> ### Author Rebuttal · Authors · 2023-08-09
>
> We thank the reviewer for the feedback and suggestions. We address here each comment/question.
>
> **Datasets for evaluation**
>
> In this work, we leverage TPVFormer for the multi-camera 2D to 3D projection. To show the effect of our contribution compared to a fully-supervised setting in a fair way, we follow the TPVFormer setup (training and evaluating on nuScenes train and val sets, respectively) and train with our self-supervised objective. The design and tuning of neural BEV projection methods (used in TPVFormer) is strongly dependent on the camera rig setup of the ego-vehicle (cameras' intrinsics, extrinsics, type and number, LiDAR density). Switching to another dataset with different sensor configurations would require specific tuning and training time. The rebuttal period was too short for such an experiment, but we would gladly consider the suggestion for extensions of this work.
>
> **Additional baselines**
>
> We thank the reviewer for the suggestion. We note that ODISE [C] (which was published and made the code available two months prior to the NeurIPs submission deadline) needs for its training the panoptic segmentation mask annotations and OVSeg [D] a segmentation model already pre-trained with segmentation mask annotations. So, both of them use a labor-intensive form of supervision that renders a direct comparison with our unsupervised POP-3D method unfair towards our work. Yet, in Tab. B of the General Response, we provide new results with ODISE. ODISE obtains 34.7 mIoU when we get 26.4 mIoU without human-label supervision. We will include this comparison in the final version of our paper. Regarding OVSeg, we did not have enough time to run it in our evaluation framework, but we would be happy to include it too in the final version of our paper.
>
> **Quantitative benchmark**
>
> We thank the reviewer for the comment. As suggested, we provide in Table A of the General Response an open-vocabulary natural language 3D grounding and retrieval benchmark, which is limited for now but which we will continue to extend. We can see that our method achieves better results than MaskCLIP+.
>
> [C] Xu et al., Open-vocabulary panoptic segmentation with text-to-image diffusion models, CVPR 2023
>
> [D] Liang et al., Open-vocabulary semantic segmentation with mask-adapted CLIP, CVPR 2023

---

> > ### Comment · Reviewer_nph2 · 2023-08-20
> > **Thank you for your answers**
> >
> > Thank you for the response. I appreciate the additional baseline results for ODISE and the commitment to add OVSeg later.
> >
> > The addition of the quantitative benchmark is a very important improvement to the paper, and it's great to see that the proposed method outperforms MaskCLIP+. I'm looking forward to seeing the completed version of the benchmark. If possible, it would be great to see the number of scenes/labeled examples for the numbers in the general response.
> >
> > Due to these added results, I will increase my rating to a weak accept.

---

### Author Rebuttal · Authors · 2023-08-09

We thank the reviewers for their valuable comments, which will allow us to improve the quality of the paper.

We are glad to note that they appreciated the clarity [*nph2, yGBj, Vrgc, q7fa*] of the manuscript, the well-defined motivation of the tackled problem [*yGBj, q7fa*], the originality [*nph2, Vrgc*] and soundness [*9pAV, Vrgc, q7fa*] of the method, its relevance [*9pAV, nph2, yGBj, Vrgc, q7fa*], and the quality of the results [*nph2, 9pAV, q7fa*].


In order to address several comments on the experimental results, we report here additional quantitative and qualitative comparisons. First, we propose a new *open-vocabulary language-driven 3D grounding and retrieval* benchmark, which we plan to further extend and present in the revised paper *[nph2-BA, 9pAV-BR, q7fa-BA]*. Second, we compare to more recent related works, as required by reviewers *[nph2, Vrcg]*, even if they require much more manual annotation than ours.


## Language-driven 3D grounding & retrieval evaluation

As suggested by the reviewers, we have built a small (due to limited time) benchmark to evaluate the "open" capability of our method quantitatively.

### The benchmark
We have collected an initial version of our *Language-driven 3D grounding & retrieval* benchmark with natural language queries. To build this benchmark, we have manually annotated 3D scenes from the validation split of the nuScenes dataset for a set of natural language open vocabulary queries. This initial set contains 7 queries, due to the time limitation of the rebuttal, but we continue to extend it. The spatial grounding for each query was obtained by manually annotating the relevant set of voxels in the scene. The objective is, given the query, to retrieve all relevant voxels in the scene. Results are evaluated using the precision-recall curve;  negative data are all the non-relevant voxels/points in the given scene. In the table below, we report the average precision (AP) for each query, further aggregated into a mean average precision (mAP) score corresponding to the mean performance across all queries. We believe this dataset, when further extended, would be an important step toward open-vocabulary analysis of driving scenes.

### How do we form the text queries
The text category descriptions used as queries in every experiment are shown in the table below. To get feature corresponding to the query text description, we follow the same approach as presented in the main paper.

| experiment name | category text description | MaskCLIP+ | POP-3D (ours) |
| ---------------:| ------------------------: | ---------:| -------------:|
|     excavator-0 | excavator                 |      14.0 |          12.8 |
|      delivery-0 | delivery vehicle          |       6.9 |           9.9 |
|      delivery-1 |                           |      15.2 |          15.5 |
|      delivery-2 |                           |      63.9 |          65.5 |
|        police-0 | police car                |      54.1 |          55.6 |
|     trash-bin-0 | trash bin                 |       9.7 |          12.6 |
|       backhoe-0 | backhoe                   |       3.3 |           4.1 |
|        **mean** |                           |  **23.9** |      **25.1** |

**Table A: AP results on our Language-driven 3D grounding & retrieval benchmark**

### Discussion about the results

We compare the performance of our method with MaskCLIP+ and report results in the table above. Our approach exhibits superior AP outcomes for 6 out of the 7 natural language queries. Overall, our POP-3D method attains a mAP of 25.1, surpassing MaskCLIP+'s mAP of 23.9.


## Comparison to more related works

We extend Fig. 4 of the main paper with more related works, which we present below in Table B. We additionally evaluate the ODISE (CVPR'23) and OpenScene (CVPR'23) methods on our task. Both methods can only be evaluated with the LiDAR-based evaluation because they do not produce 3D occupancy. We denote "2D->3D" in the table the setup where we assign 2D features to the 3D points projected into the images.

We see that both ODISE and OpenScene outperform previously reported results. However, they both use manual annotation while we don't. For instance, ODISE requires panoptic segmentation annotations for training, while OpenScene uses features from either LSeg [A] or OpenSeg [B], which are two image-language encoders that are trained with supervision. We compare here to OpenScene with OpenSeg [B] (noted "OpenScene-OS"), which requires both class-agnostic segmentation masks and image captions.

Note that we cannot evaluate MaskCLIP+, ODISE, and OpenScene for the task of image-based 3D occupancy prediction since they work either only in the 2D space (MaskCLIP+ and ODISE) or directly in the space of 3D point clouds (OpenScene), therefore do not predict the occupancy from images.

| Method                           | Type of supervision | LiDAR-based evaluation | 3D occupancy prediction from images (mIoU/IoU) |
| -------------------------------- | --------| ---------------------- | ----------------------- |
| MaskCLIP+ (2D->3D) |  none | 23.0 |      not adapted                   |
| POP-3D (ours)                    | none| 26.4                  |    16.7 / 37.7                |
| TPVFormer (supervised)           | full LiDAR segmentation | 31.3                      |  21.3 / 26.2 |
| ODISE (2D->3D)     | panoptic masks annotations | 34.7                      |        not adapted                  |
| OpenScene-OS (ensemble of 3D and 2D->3D features)                               | w. OpenSeg encoder trained w. class-agnostic seg. masks & image captions  |           38.8             |              not adapted             |

**Table B: Extension of Fig. 4 of the main paper with more related works**

[A] Li et al., Language-driven Semantic Segmentation, ICLR 2022

[B] Ghiasi et al., Scaling Open-Vocabulary Image Segmentation with Image-Level Labels, ECCV 2022

---

### Decision · Program_Chairs · 2023-09-21

**Decision:**

Accept (poster)

**Comment:**

The submission initially received mixed reviews. The authors did a good job during the discussion period, after which all reviewers became positive. The AC agrees with the recommendation. However, the authors are strongly encouraged to incorporate the feedback from the reviewers in the camera-ready version.